# Determination of the Angle of Attack on a Research Wind Turbine Rotor Blade Using Surface Pressure Measurements

Rodrigo Soto-Valle[1], Sirko Bartholomay[1], Jörg Alber[1], Marinos Manolesos[2], Christian Navid Nayeri[1], and Christian Oliver Paschereit[1]

[1]Technische Universität Berlin, Hermann-Föttinger Institut, Müller-Breslau-Straße 8,10623 Berlin, Germany
[2]College of Engineering, Swansea University, Bay Campus, Fabian Way, Swansea, SA1 8EN, UK

**Correspondence:** Rodrigo Soto-Valle (rodrigo.soto@campus.tu-berlin.de)

**Abstract.** In this paper, a method to determine the angle of attack on a wind turbine rotor blade using a chordwise pressure distribution measurement was applied. The approach used a reduced number of pressure tap data located close to the blade leading edge. The results were compared with the measurements from three external probes mounted on the blade at different radial positions and with analytical calculations. Both experimental approaches used in this study are based on the 2-D flow assumption; the pressure tap method is an application of the thin airfoil theory, while the probe method applies geometrical and induction corrections to the measurement data.

The experiments were conducted in the wind tunnel at the Hermann Föttinger Institut of the Technische Unversität Berlin. The research turbine is a three-bladed upwind horizontal axis wind turbine model with a rotor diameter of $3\ m$. The measurements were carried out at rated conditions with a tip speed ratio of $4.35$ and different yaw and pitch angles were tested in order to compare the approaches over a wide range of conditions.

Results show that the pressure tap method is suitable and provides a similar angle of attack to the external probe measurements as well as the analytical calculations. This is a significant step for the experimental determination of the local angle of attack, as it eliminates the need for external probes, which affect the flow over the blade and require additional calibration.

## 1   Introduction

The angle of attack (AoA) is, by definition, a 2-D concept. Nevertheless, on a wind turbine, the rotating system, i.e. a blade, is under 3-D effects such as tip and root vortices, yaw misalignment, velocity inductions, among others that render the precise determination of the AoA difficult (Shen et al., 2009). Additionally, the AoA is indirectly obtained through pressure or velocity fields, thus several uncertainties are added in its estimation. In this way, determining the local AoA on wind turbine blades remains one of the greatest aerodynamic challenges. At the same time, the determination of AoA is necessary in order to calculate lift and drag forces over the blade, develop accurate aeroelastic models, or establish a control tool.

The AoA can be calculated according to its geometrical definition using the velocity triangle defined by the wind velocity and the rotational speed. Unfortunately, this estimation relies on well known freestream conditions and does not take into

account induction effects. Therefore, if a more reliable estimation is required, it is necessary to use on-blade measurement tools.

Most of the on-blade measurements use external probes to measure the local pressure. Various methods have been used, while follow the same principle: apply a correction due to the upwash induced by the presence of the blade itself. Including a stagnation pressure hole leaves the 3-hole probe as required minimum. Additional holes (5-, 6-, 7-) allow the cross flow derivation and provide better accuracy. However, the number of calibration curves increases, thus the determination of the inflow becomes more difficult (Schepers and Van Rooij, 2005).

Several field measurements have been conducted using probes as one of the estimation methods for the AoA. Brand et al. (1997); Simms et al. (1999); Madsen et al. (1998); Maeda et al. (2005); Bak et al. (2011a) showed measurements results employing 5-hole probes from the Energy research Centre of Netherlands (ECN), The National Renewable Energy Laboratory (NREL), Technical University of Denmark (DTU), Mie University (Mie) and DanAero projects, respectively (see Table 1). Bruining and van Rooij (1997) used 3-hole probes in the Delft University of Technology (DUT) project. The upwash correc-
tion was made based on wind tunnel measurement of static blade or airfoils representative of the studied blade section. It is remarkable that the case of the ECN exhibited better results without the upwash correction. This is assumed as the compensation effect of the downwash from the shed vorticity due to the variation of the bound circulation along the blade span (Schepers et al., 2002).

    These methodologies have been applied over wind turbine models on tunnel experiments. Gallant and Johnson (2016)
presented the determination of the AoA using a 5-hole probe on a three bladed turbine model at the University of Waterloo (UW) wind tunnel facilities. A combination of geometrical and induction corrections, based on the work of Hand et al. (2001), was applied to obtain the AoA for different yaw offsets and tip speed ratios. The results show a good trend agreement between the probe measurements and the model proposed by Morote (2016). The operation range of the 5-hole probe was studied by Moscardi and Johnson (2016) for a large range of pitch and yaw angles ($\pm50°$), using the test rig with only one blade.

Bartholomay et al. (2018) showed AoA estimation through 3-hole probes, from the Berlin Research Turbine (BeRT). The 3-hole probe calibration was made under axial inflow and performed on-blade operation for axial and yawed inflows up to $30°$. The results showed a good agreement with CFD computations (Klein et al., 2018) under the same operation points.

    In general, according to the published literature, external probes can be used to determine the AoA. However, in the case of wind turbine models, such probes are intrusive and significantly disturb the flow over the blade section where they are mounted.

Other complementary tools, used on research turbines are surface pressure sensors, located along the blade chord. These sensors are used to record the pressure distribution along the blade chord at a desired radial position and to calculate the aerodynamic loads. Different computational methods use this information as a source to estimate the AoA.

    The inverse BEM method is probably the most common. From the surface pressure sensors, the normal and tangential forces are calculated. Assuming that they are uniform over an annulus containing the blade section. The wake-induced velocities
are calculated according to momentum theory, yielding the effective velocity vector and subsequently the AoA (Whale et al., 1999). This method was implemented by ECN, NREL, DTU projects, obtaining similar results with their respective estimations based on probes.

The NREL suggested an algorithm to estimate the AoA from pressure distribution values under axial (Sant et al., 2006a), unsteady (Sant et al., 2006b) and yawed conditions (Sant et al., 2009). The method assumes an initial AoA distribution. The lift is then calculated for each azimuth and radial position based on the pressure surface data and the AoA. Afterwards, the bound circulations were determined by means of the Kutta–Joukowksi theorem for a lifting line. The resulting values were prescribed in a free wake vortex model to obtain a new AoA based on the induced velocities to finally iterate until the AoA converged.

Schepers et al. (2012) presented the inverse free wake method applied to the MEXICO rotor, which follows the same BEM principle but using the normal and tangential forces into a free wake model. Several computational methods can be found in the latest phase of the project, summarized by Schepers et al. (2018), such as azimuth average, three point and lifting line average methods among others.

The surface pressure measurements also allow experimental estimations. Shipley et al. (1995) showed the stagnation point normalization method described as follows: the local dynamic pressure is estimated as the maximum value of the pressure side in each pressure distribution station. This value is used to estimated the freestream velocity and then the AoA based on the geometrical velocities defined by pitch, yaw and azimuth angles.

Moreover, Brand (1994) presented the stagnation point method. The AoA is estimated as follows: The stagnation point is located as the previous method. Afterwards, the intersection of the chord line and a line normal to the surface at the stagnation point is used to estimate AoA. The position of the point of intersection can be determined 2D approaches either codes or wind tunnel measurement (Whale et al., 1999). The drawback of this method is that it relies only in the geometry of the blade section, assuming AoA and Reynolds number no influence.

Furthermore, Bruining and van Rooij (1997) exposed an additional method that use two frontal pressure taps, one on the pressure side and one on the suction side, working as an built-in probe in the blade. The drawback of this is that requires calibrating the blade station where the taps are located.

Schepers et al. (2002) reported the comparison between experimental probes, pressure taps and inverse BEM methods regarding the field measurement from ECN, NREL, DUT, DTU and Mie. The main conclusions found were: (1) The ambiguity of the 3D AoA definition implies that any check on accuracy can only be carried out with an arbitrary reference. (2) Before stall, the estimations of the AoA remain with differences below $1°$. (3) Above stall conditions, the differences between methods can go up $4°$. Table 1 shows field and wind tunnel experiments with the most common estimation methods mentioned above.

Therefore, the pressure distribution over a rotating section can be used to relate the AoA, if it is comparable with nonrotating conditions, where the AoA is known. Several investigations showed a relation between 2-D and 3-D pressure distribution. Ronsten (1992) showed a good agreement between the pressure distribution over nonrotating and rotating blades along span positions of $r/R \geq 0.55$ and $r/R \geq 0.3$ at tip speed ratio of $4.32$ and $7.37$, respectively.

Guntur and Sørensen (2012) presented different methods to determine the AoA for the MEXICO rotor (Bechmann et al., 2011) based on CFD data. One of the approaches is based on matching up $C_P$ distributions from 2-D and 3-D data, where the AoA was known in the former case. This method has a good agreement for small angles of attack ($< 10°$) and in the middle blade region ($0.25 \leq r/R \leq 0.85$). The latter points out an alternative method to estimate the AoA where the 2-D and 3-D pressure distribution are comparable.

**Table 1.** Angle of attack estimation methods on wind turbine rotor blades.

| Contributor | Blades | Radius [m] | $Re_c^a$ | On-blade tool | Estimation method |
|---|---|---|---|---|---|
| **Field** | | | | | |
| ECN[b], Brand et al. (1997) | 2 | 13.72 | 1.8M[c] | 5-hole probe, pressure taps | stagnation point, probe measurements, inverse BEM |
| DUT[b], Bruining and van Rooij (1997) | 2 | 5 | 0.9M[c] | 3-hole probe, pressure taps | inverse BEM, stagnation point, probe measurements, frontal pressure taps |
| NREL[b], Simms et al. (1999) | 3 | 5 | 0.7M[c] | wind vane, 5-hole probe, pressure taps | probe measurements, stagnation point normalization, matching up $C_P$, inverse BEM |
| DTU[b], Madsen et al. (1998) | 3 | 9.5 | 1M[c] | 5-hole probe | probe measurements |
| MIE[b], Maeda et al. (2005) | 3 | 5 | 0.5M[c] | 5-hole probe, pressure taps | probe measurements |
| DanAero, (Bak et al., 2011a) | 3 | 40 | $1.5 - 6.1$M | 5-hole probe, pressure taps, microphones | probe measurements, matching up $C_P$ |
| **Wind Tunnel** | | | | | |
| MEXICO, Schepers et al. (2012) | 3 | 2.25 | 0.8M[d] | pressure taps | inverse BEM, inverse free wake, based on CFD |
| LMEE, Sicot et al. (2008) | 2 | 0.67 | 300k | pressure taps | lifting line |
| BeRT, Klein et al. (2018) | 3 | 1.5 | 290k | 3-hole probe, pressure taps | probe measurements, based on CFD |
| UW, Moscardi and Johnson (2016) | 3 | 1.7 | 300k | 5-hole probe | probe measurements |

(a) $Re_c$: Reynolds number based on chord length at 70%R and relative inflow velocity. (b) Additional information can be found on the International Energy Agency (IEA) Annexes reported by Schepers et al. (1997) and Schepers et al. (2002). (c) Summarized in the IEA Annexes reported by Schepers et al. (2002). (d) Reported by Schepers and Schreck (2019).

Maeda et al. (2005) showed surface pressure comparison between field measurements and wind tunnel experiments. The latter was carried out using the same blade in stationary conditions. A good agreement was shown, regarding the surface pressure distribution under prestall (AoA= 10°) and stall (AoA= 16°) conditions. In the case of a poststall (AoA= 20°) condition, the results of the wind tunnel present a reduced pressure magnitude on the suction side, in contrast with the field case.

Bak et al. (2011b) studied the pressure distribution on a wind turbine in atmospheric conditions and in a wind tunnel. The wind tunnel experiments were carried out with 2-D wing, taking the characteristics of four specific sections from the turbine. The agreement remains valid for small angles of attack ($< 12°$) and for the outer region of the blade ($r/R > 0.4$).

Overall, it is generally agreed that static 2-D wings and rotating blades have a good agreement in surface pressure measurements, at least for attached flow conditions. This opens the possibility of using methods based on the blade chord pressure distribution to estimate the AoA, in the range of agreement.

Gaunaa (2006) developed an analytical solution for the unsteady 2-D pressure distribution on a variable geometry airfoil undergoing arbitrary motion, based on thin airfoil theory. Further investigations made by Gaunaa and Andersen (2009), using this method, related the pressure over the airfoil with the effective AoA. The added benefit of the specific method is its simplicity, as it only requires the pressure difference between the airfoil pressure and suction side at one or two chordwise positions and at the same time can be performed while operating in unsteady conditions.

To the authors' knowledge, this method has not been applied on a rotating blade yet. Given the good agreement between 2-$D$ and 3-$D$ pressure distributions away from the root region, this paper presents an alternative method of determining the AoA by means of pressure tap measurements. The present investigation aims at providing experimental verification for one such surface pressure method (Gaunaa and Andersen, 2009) on the rotating blade.

Nowadays, new technologies such as passive fiber optic pressure sensors presented by Schmid (2017) able to perform quasistatic and unsteady measurements of rotor blades in operation that can withstand harsh conditions. Therefore, the development of new methods to determine the AoA based on pressure distribution data would provide valuable information without the necessity of invasive tools.

The Technical University of Berlin has developed a scaled wind turbine model, BeRT, equipped with 3-hole probes and pressure taps on one of its blades (Vey et al., 2015). The results presented here are the first on-blade pressure measurements from the BeRT blade and can be used to validate numerical solvers and to develop future control strategies.

In the remaining of the paper, the facilities and the research turbine model are described, followed by the methodology to determine the AoA and to assess the validity of the Gaunaa method on the rotating plane. The results are presented in Sect. 4 and the paper closes with concluding remarks in Sect. 5.

## 2 Experimental setup

### 2.1 Wind tunnel

The tests were conducted at the Hermann Föttinger Institut of the Technische Universität Berlin in the GroWiKa (large wind tunnel), a closed-loop wind tunnel driven by a $450\ kW$ fan and a cross-sectional area $A_{tunnel} = 4.2 \times 4.2\ m^2$ presented in Fig. 1 (left). The turbine model was placed at the large test section, where the maximum velocity is $10\ \mathrm{ms}^{-1}$. The setup was reproduced from the work of Bartholomay et al. (2017), in which the flow quality was measured and the reproducibility of the flow was evaluated. In order to keep the turbulence intensity on a comparable level, one homogeneous filter mat and three screens were positioned in the crosssections upstream of the turbine as it can be seen in Fig. 1 (left). The turbulence intensity achieved with this setup is less than $1.5\%$. With this level of turbulence, it can be expected small variations between rotations of the turbine, that suggests using multiple rotations to achieve a significant statistical average in the data.

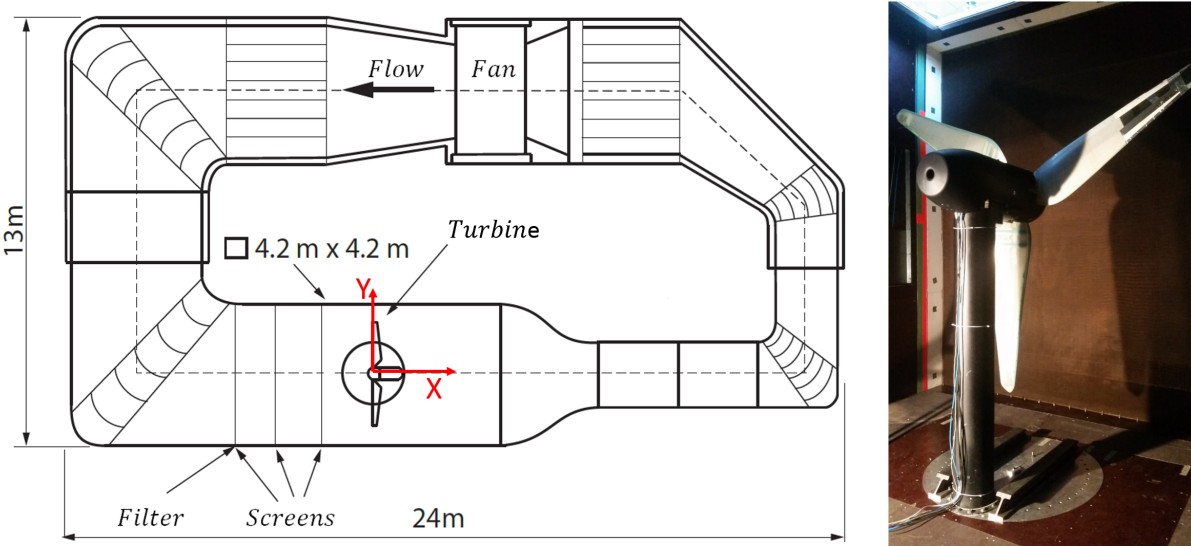

**Figure 1.** Outline of GroWiKa, modified from Klein et al. (2018) (left). Berlin Research Turbine - BeRT in the wind box (right).

At the same time, the inflow showed some heterogeneity, i.e. was not fully uniform as is depicted in Fig. 2 (left). Figure 2 (right) shows four axial velocity distributions over at the radial positions 45, 65, 75 and $85\%R$. Therefore, due to this characteristics it was decided to analyze the measurement data over small azimuth angle stations.

Additionally, the dynamic pressure is monitored by two Prandtl tubes located at the walls at $0.43R$ upstream the turbine at $2.7\ m$ height. Based on the Prandtl tubes, all test cases were conducted with a freestream velocity of $U_\infty \approx 6.5\ \mathrm{ms^{-1}}$.

## 2.2    Wind turbine model

BeRT, Fig. 1 (right), is a three-bladed upwind horizontal wind turbine with a rotor radius of $R = 1.5\ m$. The turbine yaw angle and the blade pitch angle were fixed during the measurements. Figure 3 (left) shows a reference sketch for the azimuth ($\phi$) and
yaw ($\psi$) angles.

A slightly modified Clark-Y airfoil profile is used along the entire blade span and there is no cylindrical root section. The airfoil modification was necessary in order to account for a realistic trailing edge thickness with respect to manufacturing requirements. Aerodynamically, the design intended to avoid stall while keep offering optimal performance and the maximum internal space to include instrumentation (Pechlivanoglou et al., 2015).

In this way, the specific airfoil profile was chosen as it performs well at low Reynolds number ($Re$), i.e. at the conditions relevant to BeRT ($Re$ range of $170 - 300k$ along the span). The blade twist was selected so that the local AoA stays constant over the span at rated conditions. Figure 3 (right) illustrates the definition of the main angles and velocities over a blade section and Fig. 4 (left) shows the twist and chord distributions.

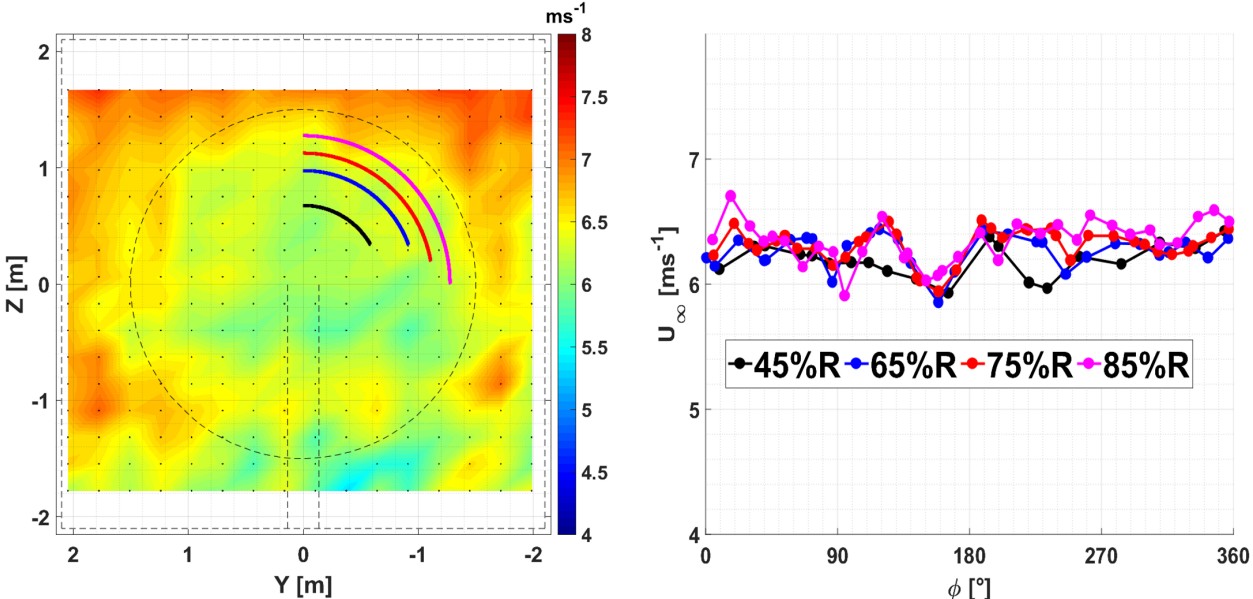

**Figure 2.** Axial inflow. Dashed lines: tip and tower positions. Colored lines: radial positions at 45, 65, 75 and 85%$R$ following the blade rotation (left). Velocity distributions over radial positions at 45, 65, 75 and 85%$R$ (right).

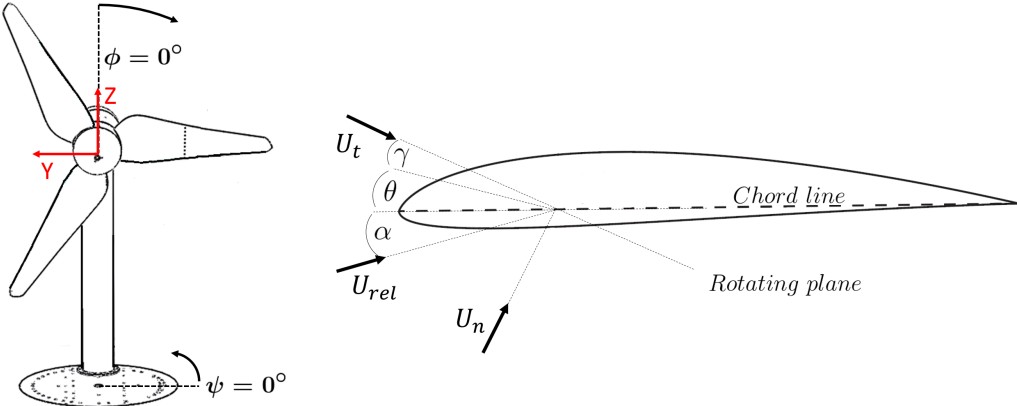

**Figure 3.** Angles definition. Azimuth, $\phi$ and yaw, $\psi$ (left). Angle of attack, $\alpha$, pitch, $\theta$ and twist, $\gamma$. $U_t$, $U_n$ and $U_{rel}$ are the tangential, normal and relative velocities, respectively (right).

The turbine rotor area ($A_{BeRT}$) produces a considerable blockage ratio in the wind tunnel, $\epsilon = A_{BeRT}/A_{tunnel} \approx 0.4$. The blockage effect was analyzed in terms of the equivalent freestream velocity ($U'$) which produces the same torque. Glauert (1926) showed that for a propeller the ratio between the wind tunnel velocity ($U_\infty$) and its corresponding equivalent freestream velocity is a function of the blockage ratio and the thrust coefficient($C_T$), Eq. 1. Using the BeRT rotor characteristics reported

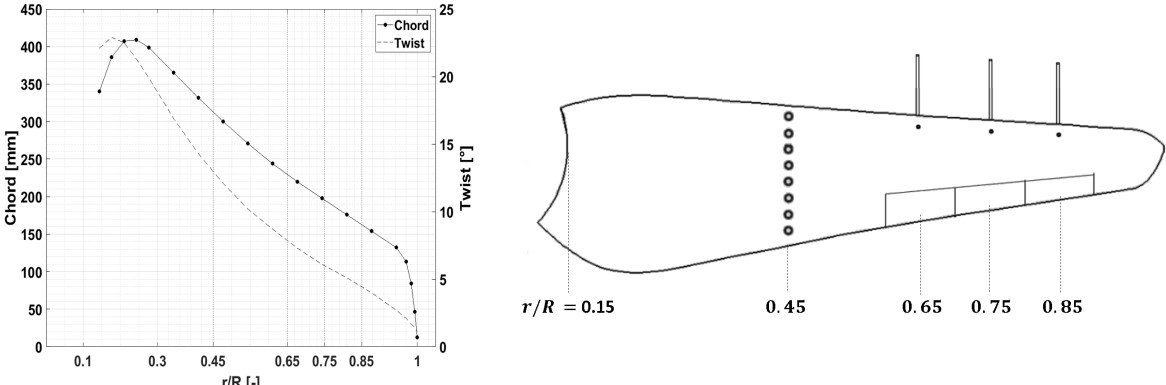

**Figure 4.** Twist and chord distribution along span (left). The rotor blade with 3-hole probes and pressure taps over span position (right).

by Marten et al. (2019), a thrust coefficient of $C_T = 0.77$ (expected at rated condition) was considered. Subsequently, applying Eq. 1, implemented on wind turbines, results in the velocity ratio of $U_\infty/U' = 0.86$.

$$\frac{U_\infty}{U'} = \left(1 - \left(\frac{\epsilon C_T}{4\sqrt{1+C_T}}\right)\right)^{-1} \tag{1}$$

It is noted that this correction has also been applied successfully in wind tunnel experiments with even higher blockage ratio (45%, Refan and Hangan (2012)).

One blade was equipped with pressure taps and three 3-hole probes at different radial positions, as shown in Fig. 4 (right). Due to manufacturing reasons (internal structure, holes spacing), the pressure taps could only be located at a single spanwise location, which was at 45% of the blade span. Each pressure tap was connected through silicone tubes inside the blade to a pressure box located in the hub which contains all sensors. The average length for the tubes between tap and sensor was $650mm$ which included an arrangement between cannulas and tubes as shown in Fig. 5.

The 3-hole probes were located at 65, 75 and $85\%R$ and mounted on the pressure side (see Fig. 6, left). The 3-hole probes consist of one straight tube in the middle, accompanied by two outer tubes with a $45°$ nozzle (see Fig. 6, right). Each outer tube was connected to a differential pressure sensor through a silicone tube, using the middle one as a reference. The sensors were installed at the spanwise position of each probe, reducing the tube length to less than $100$ mm.

All pressure transducers were installed in such a way that their membranes were parallel to the plane of rotation to minimize the centrifugal effect on them. More information about the sensors can be found in previous work by Vey et al. (2015), while the calibration and data acquisition procedure is detailed in the Sect. 3.1.

The blade was also provided with three trailing edge flaps with $10\%R$ span length and $30\%c$ chord length and located consecutively from $60\%$ to $90\%$ along the span. Each 3-hole probe was aimed to give feedback information to choose flap movements. However, The flaps were fixed without any deflection for all test cases presented in this study. The turbulence transition was not fixed over the blades, in contrast to the previous work of Klein et al. (2018).

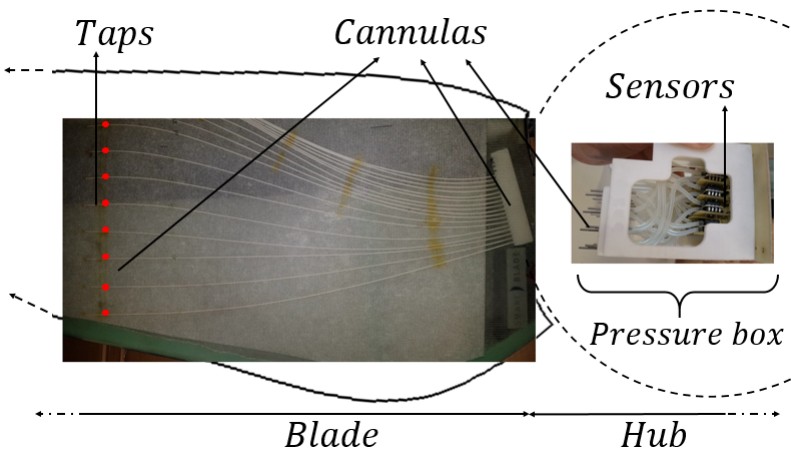

**Figure 5.** Tubing details between pressure taps and sensors.

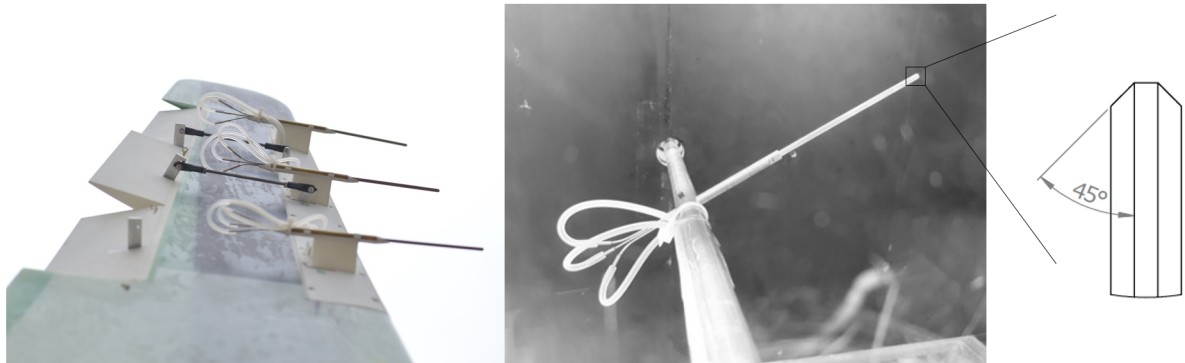

**Figure 6.** 3-hole probes mounted in the equipped blade (left). Calibration of a 3-hole probe and tip details (right). It is noted that although the flaps appear deflected in the photo, they were always in the neutral position for the experiments of this campaign.

Rotating (NI cRIO 9068) and nonrotating (NI cDAQ 9188) measurement systems were synchronized and located in the hub and the external control cabinet, respectively. The measurement data were recorded using NI 9220 modules with an acquisition frequency of $10\ kHz$.

The pressure data from the blade were recorded through the rotating system, while the freestream dynamic pressure through the nonrotating system. The blade position was recorded through a Hall effect sensor located in the nacelle. Each measurement was recorded and phase averaged until 100 rotations were completed, with an azimuth step of $\Delta\phi = 1°$.

 **3   Methodology**

In this section, the methodology of this research is described. The main idea is to compare the results obtained by the method proposed by Gaunaa and Andersen (2009) when it is applied to the pressure tap data against the AoA from the 3-hole probe measurements and analytical calculations.

According to the BeRT design specification, the combination of chord and twist distribution achieves an optimal shape
 (Pechlivanoglou et al., 2015) which provides a constant AoA over most of the blade span (Bartholomay et al., 2017), so the AoA at the radial position of the pressure taps and the 3-hole probes should be the same under aligned flow conditions.

The calibration of the sensors, the applied corrections and the description of the methods used to determine the AoA follow, while the test cases and their uncertainty are summarized at the end of this section.

**3.1   Calibration**

 Differential pressure sensors were used for both experimental methods, the pressure taps ($HCL0025E$) and the 3-hole probes ($HCL0075E$). During the calibration of the sensors, the turbine was in a static position and a constant pressure was provided to achieve eleven calibration pressure points using the external calibrator, Halstrup KAL 84. All calibrations were linear and the fitting curves showed a coefficient of determination values of $R^2 \geq 0.999$.

The 3-hole probes were calibrated in a small wind tunnel. The calibration range was from $-30°$ to $30°$ with steps of $0.5°$.
 The calibration was made between the normalized pressure and the swept angles following the standard procedure described by Dudzinski and Krause (1969). Subsequently, the calibration was repeated for inflow velocities from 16 to 22 ms$^{-1}$ with steps of $\Delta U = 2$ ms$^{-1}$. The velocity range was selected so that it covers the relative velocity perceived by the blade in the range $0.45 \leq r/R \leq 0.85$, i.e. the location of the 3-hole probes. The AoA fit remains linear within $-10$ to $10°$, getting a nonlinear fit for larger angles.

 **3.2   Pressure correction**

The pressure sensors measure the differential pressure ($P_{si}$). The 3-hole probes use the inner tube as a reference, while the pressure taps use the static pressure in the test section.

The structural design of BeRT results in eigenfrequencies of the blades of $f_{blade} \geq 13.5 Hz$ and the tower $f_{tower} \geq 18 Hz$. For this reason, the data were lowpass-filtered using a Butterworth filter with a cut off frequency of 12Hz to reduce the noise
 and structural vibrations. Figure 7 shows the raw signal spectra over one 3-hole probe pressure sensor at $75\% R$ and the pressure tap at $x = 2\% c$. It can be seen that the main variations are influenced by the rotational frequency of 3Hz and its harmonics.

The dynamic response of the taps/tubes system was evaluated theoretically following the model proposed by Bergh and Tijdeman (1965). Figure 8 (left) shows a scheme of the model used to apply the analysis, based on the tube arrangement depicted in Fig. 5, while Fig. 8 (right) shows the theoretical response of the system, based on Bergh and Tijdeman (1965).
 In order to minimize the attenuation and phase lag of the signal, an additional low pass filter was applied, with a cut off

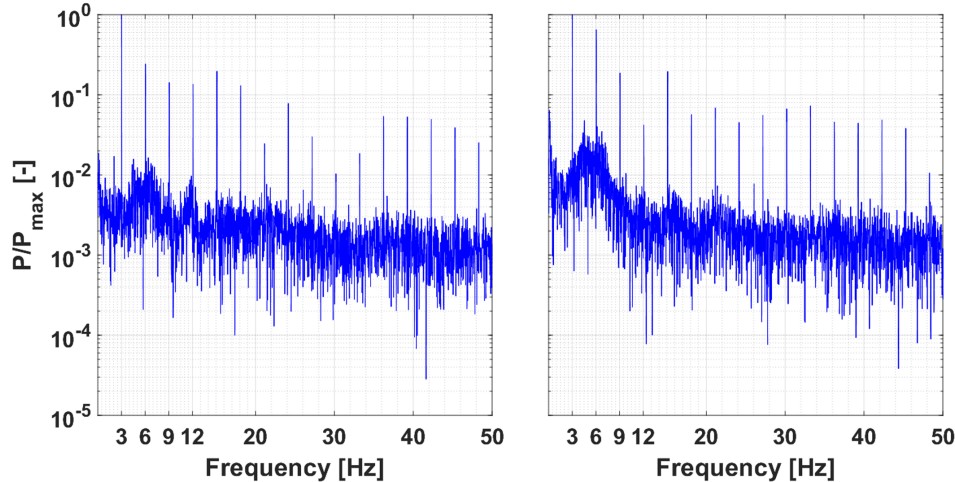

**Figure 7.** Frequency spectrum of one pressure sensor of the 3-hole at $75\%R$ (left). Frequency spectrum of the pressure tap at $x = 2\%c$ (right). Both cases on pitch angle $\theta = 0°$ and yaw angle $0°$

frequency of 6Hz. This was considered adequate as it shows the amplitude amplification and phase lag are less than $1\%$ and $10°$, respectively.

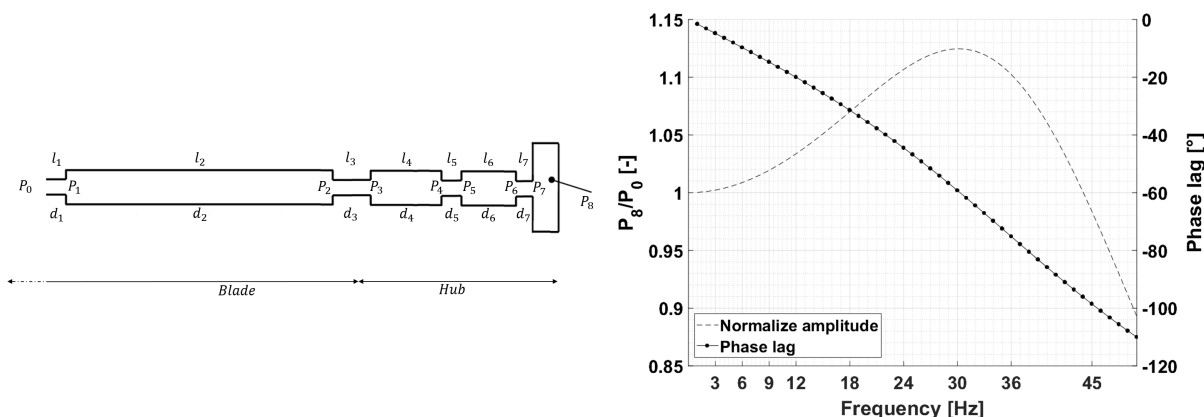

**Figure 8.** Scheme of the model to apply Bergh and Tijdeman (1965) dynamic response analysis, $P$, $l$ and $d$ are the pressure, length and diameter of each section (left). Theoretical dynamic response of the amplitude and phase lag (right).

In the case of the pressure taps, the centrifugal effect was quantified and corrected, Eq. 2, based on Hand et al. (2001), where $r_i$ is the radial position of the pressure tap $i$ and $\Omega$ is the turbine angular velocity, $2\pi f$.

$$P_{corr} = P_{si} + \frac{\rho}{2}(\Omega r_i)^2. \tag{2}$$

The hydrostatic correction has less impact since all the sensors are located in the hub, and was consequently neglected.

## 3.3 Methods to determine the angle of attack

### 3.3.1 3-hole probes

The method to determine the AoA from the 3-hole probes was based on previous work with the same setup. It is outlined here for completeness, while further details can be found in Bartholomay et al. (2017). Figure 9 shows the reference system for an arbitrary blade section, with 3-hole probe installed.

The AoA relative to the probe, $\alpha_{probe}$, was identified from the 3-hole probe calibration, through their normalized pressure, Eq. 3, where $P_1$ and $P_2$ are the outer tubes, $P_0$ the reference tube and $\overline{P}$ the average between the outer tubes.

$$C_{P,\ probe} = \frac{P_1 - P_2}{P_0 - \overline{P}}.$$  (3)

However, as shown in Fig. 9, a geometrical rotation between the probe and the section coordinates was necessary to evaluate the AoA in the respective blade section, $\alpha_{probe,section}$. The latter angle differs from $\alpha$, which is the effective AoA of the blade section, because the blade itself induces a velocity on its surroundings. To correct this, XFOIL (Drela and Youngren, 2001) calculations were used to estimate the velocity at the probe location, under the assumption of 2-D flow. Afterwards, a fit function was found between the effective AoA, $\alpha$, and $\alpha_{probe,section}$. Equation 4 shows an approximation of the downwash correction (Klein et al., 2018).

$$\alpha = 0.58° \alpha_{probe} - 0.64°$$  (4)

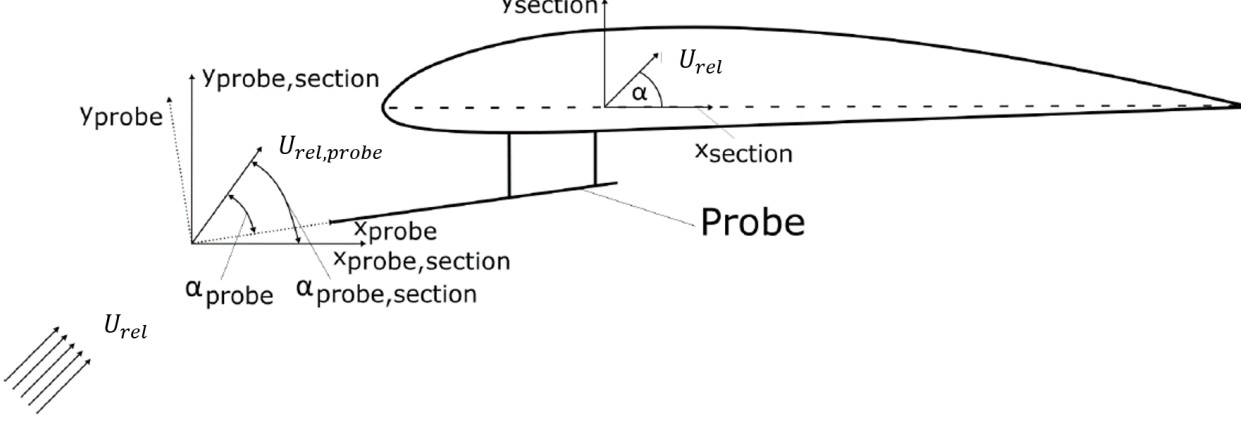

**Figure 9.** Schematic of the reference system for a probe, modified from Klein et al. (2018).

As the turbine was set under yaw misalignments, it is important to verify the effectiveness of the 2D probe. The range of the AoA, in the probe stations, is $0° \leq \alpha \leq 10°$. Therefore, adding the corresponding twist angle, the range of the AoA relative to

the probes, $\alpha_{probe} \leq 18°$. Moreover, the probes are aligned with the chord, thus the yaw angle relative to the probe is the same $-30° \leq \psi_{probe} \leq 0°$.

Zilliac (1993) and Moscardi and Johnson (2016) determined the mono-zone as $\pm 30°$ ($\alpha_{probe}, \psi_{probe}$). This zone represents where the calibration parameters of the probes remain invariant, i.e. $C_{P, \, probe}$. These studies used probes with 7- and 5-holes, respectively. As a 3-hole probe sweep the same angle of these calibrations, its mono-zone should be the same.

Moreover, Bruining and van Rooij (1997) employed 3-hole probes on field measurements with good agreement of the AOA, compared to inverse BEM and stagnation point methods. In addition, Klein et al. (2018) showed similar results from experimental and CFD simulations where the wind tunnel structure was considered. Therefore, based on these arguments, it was assumed that the 3-hole probes are able to estimate the AoA in the yaw misalignments here studied.

### 3.3.2 Pressure taps

The determination of the AoA from the pressure distribution on the blade section was based on the unsteady model developed by Gaunaa (2006). The main assumptions for this methodology rely on the thin airfoil theory and low Mach number. This allows modeling the airfoil as its camberline together with the assumptions of inviscid, incompressible, and irrotational flow.

Aiming at simpler solutions to estimate airfoil loads that can be applied on active load control, and based on the considerations mentioned above Gaunaa (2006) formulated an analytical expression for the forces over an arbitrary airfoil shape. This expression relates the pressure difference between the lower and upper side, over the camberline, with the velocity potential field, aerodynamic forces, and pitching moment. Gaunaa and Andersen (2009) summarized this formulation in Eq. 5, as the normalized pressure and its contributions, where $\Delta P(\mathbf{x})$ is the pressure difference between the lower and upper side at a specific chordwise position and $q = 0.5\rho U^2$ is the dynamic pressure.

$$\frac{\Delta P(\mathbf{x})}{q} = g_c(\mathbf{x})\alpha_{c,eff} + g_{camb}(\mathbf{x}) + g_{\dot{\alpha}}(x)\frac{\dot{\alpha}c}{U} + g_\beta(\mathbf{x})\beta + g_L(\ddot{y}, \ddot{\alpha}, \dot{\beta}, \ddot{\beta}, \mathbf{x}). \tag{5}$$

It is important to note that this summary neglects the chord streamwise degree of freedom, i.e. $\dot{X} = \ddot{X} = 0$.

On the right side of Eq. 5, $g_c(\mathbf{x})$ corresponds to the influence of the circulatory forces. This contribution is modulated by $\alpha_{c,eff}$, the effective AoA that takes into account the time lag effects caused by the vorticity shed into the wake, for simplicity, now considered as $\alpha$.

The remaining contributions in Eq. 5 depend on the instantaneous motion of the airfoil, known as added mass terms. The second and third terms, $g_{camb}$ and $g_{\dot{\alpha}}$ correspond to the added mass due to the basic camber line and pitching, respectively.

The formulation allows the calculation of the effect of a flap on the airfoil, with $\beta$ being the flap angle. This contribution in the model is considered with the added mass term $g_\beta$. Since there is no flap at the $45\%$ span position, the flap deflection angle is set to $\beta = 0°$ and therefore $g_\beta$ is eliminated.

The term $g_L$ contains the nonlinear contributions. Gaunaa and Andersen (2009) claim that the addition of the geometrical nonlinearities does not change the conclusions from linear estimation for the most part of the chord, except for a zone very close to the leading edge. Based on this consideration, the term $g_L$ is neglected.

Gaunaa and Andersen (2009) and Velte et al. (2012) suggested a control variable based only on two pressure taps. To achieve this, the contribution of the pitching added mass term, $g_{\dot{\alpha}}$ was neglected by choosing a specific chord position where its value is zero.

Equation 6 shows the reduced relation between pressure distribution and AoA, where $k_1 = g_c(\mathbf{x} = 0.125)$ and $k_2 = g_{camber}(\mathbf{x} = 0.125)$. An extended review of the two dimensional theory and the mathematical derivation of this method and applications, can be found in Gaunaa (2002, 2006).

$$\frac{\Delta P(0.125)}{q} = k_1\alpha + k_2 \implies \alpha = \frac{1}{k_1}\left(\frac{\Delta P(0.125)}{q} - k_2\right). \tag{6}$$

Several studies made by Gaunaa (2002); Gaunaa and Andersen (2009); Velte et al. (2012), related the same theory on wing experiments and computational models, with a Risø-B1-18 and NACA64418. Thus, it is assumed that the linearity, applied on the remaining terms, is a good approximation for a Clark Y airfoil shape, which is thinner (11.8%) than the other airfoils where the method was successfully applied.

In order to obtain the constants $k_1$ and $k_2$ from Eq. 6, XFOIL calculations were computed. The AoA was swept from $-3°$ to $10°$. The Reynolds number ($2.5 \times 10^5 \leq Re \leq 3.0 \times 10^5$) and free transition method ($4 \leq NCrit \leq 12$) influence were studied with no significant changes. Subsequently, a linear curve fit was made between normalized pressure ($\Delta C_P(0.125)$) and the AoA swept. The fit values are $k_1 = 0.23$ and $k_2 = 0.43$, with a coefficient of determination of $R^2 \geq 0.999$.

Finally the AoA was calculated using Eq. 6, where $\Delta P(0.125) = P_{lower}(0.125) - P_{upper}(0.125)$.

Figure 10 shows a good agreement between the pressure distribution from the rotating blade and the computational tool in the estimated angle. The difference between both curves is $\Delta C_P \leq 0.05$ until $x = 30\%c$, except the peak at the suction side, $\Delta C_P(x = 1\%c) = 0.2$. Afterwards, $\Delta C_P$ varies between $0.05 - 0.10$. This agrees with the fact that rotation does not have a great impact over the pressure distribution in the attached flow operations points (Ronsten, 1992; Corten, 2001).

Since there are no pressure taps in the exact $12.5\%c$ position, a linear interpolation was made, between $[10 - 15]\%c$ for the suction side and $[10 - 30]\%c$ for the pressure side.

The relative dynamic pressure, $q_{rel} = 0.5\rho U_{rel}^2$, was considered equal to the maximum value in pressure side distribution, i.e. at the stagnation point (Shipley et al., 1995), for each azimuth station. This was required for the yaw misalignment cases, where the dynamic pressure is variable with azimuth position.

### 3.3.3 Analytic estimation

The introduction of a yaw misalignment produces an expected change in the AoA distribution along the blade span due to the crossflow i.e. depends on the azimuth angle variations. Therefore, a geometrical approach was used to compare the experimental methods under these operational points, as pressure taps and 3-hole probes location differs in radial position.

The normal velocity contribution is a function of the yaw angle, Eq. 7. On the contrary, the tangential velocity contribution depends on the rotational speed, yaw and azimuth angle, Eq. 8, due to the crossflow presented (see Fig. 3). Using these geometrical velocities contribution and the axial, $a$, and tangential, $a'$, factors simulated with the BEM-module QBlade (Marten

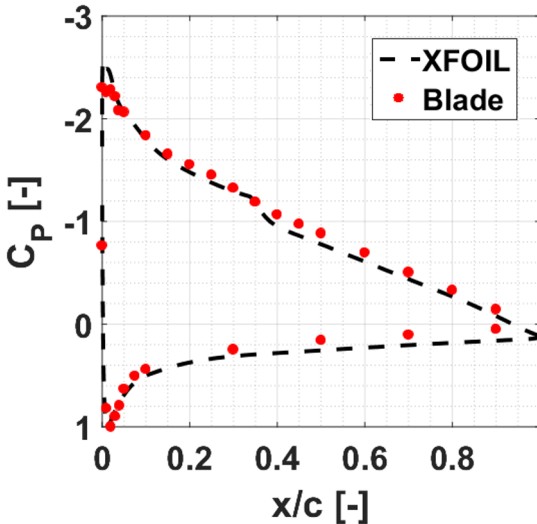

**Figure 10.** XFOIL ($\alpha = 7.6°$) and measured pressure distribution of the current setup at yaw angle of $\psi = 0°$, pitch angle of $\theta = 0°$ and azimuth angle of $\phi = 0°$.

et al., 2015) an analytical AoA was estimated as is shown in Eq. 9.

$$U_n = U_\infty cos(\psi) \qquad (7)$$

$$U_t = \Omega r - U_\infty sin(\psi)cos(\phi) \qquad (8)$$

$$\alpha_{geo} = atan\left(\frac{U_n(1-a)}{U_t(1+a')}\right) - \theta - \gamma \qquad (9)$$

The blockage effect must be considered. Consequentely, the inflow velocity ($U_\infty$) for these calculations was replaced by the equivalent freestream velocity. Thus, applying Eq. 1 results in the equivalent freestream velocity of $U' = 7.5ms^{-1}$.

Equation 9 can be used to estimate the AoA in the aligned case, which is independent of the azimuth angle, as the yaw angle is zero. Therefore, the AoAs have small variations, regarding the induction factors. Thus, the AOA in the location of the pressure taps and 3-hole probes takes the value of $\alpha_{geo,\psi=0°} \approx 6.7°$, when the pitch angle is set at $\theta = 0°$.

### 3.4 Test cases and measurement uncertainty

Several operational conditions were analyzed, three yaw angles $\psi = 0°$, $-15°$, and $-30°$, and for each yaw angle, the pitch angle was swept from $-2°$ to $6°$ in steps of $\Delta\theta = 2°$. For all cases, the tip speed ratio was fixed $\lambda = 4.35$.

The measurement uncertainty, for all quantities, was taken into account in order to quantify the error magnitude over the results. Both AoA estimation approaches have the same iteration in the error propagation, based in the following steps:

1. Nominal error of each sensor.

2. The standard deviation of the averaged measurements. This was calculated with the same azimuth step as the phase average.

3. Conversion to AoA. Thus, the error propagation after applying Eqs. 3 and 6 for the 3-hole probes and pressure taps, respectively.

Table 2 shows the overall uncertainty for all the quantities. The point 3. depends highly on the values of the measured pressure. For this reason, Table 2 shows the minimum and maximum values. An example of the uncertainty over the azimuth angle of each tool can be seen in App. D1.

**Table 2.** Measurement uncertainty summary.

| Measurement | Uncertainty | Range |
|---|---|---|
| Yaw angle, $\Psi$ | $\pm 0.5°$ | $\pm 30°$ |
| Pitch angle, $\theta$ | $\pm 0.5°$ | $\pm 15°$ |
| Azimuth angle, $\phi$ | $\pm 0.5$ | $0°$ to $360°$ |
| Dynamic pressure | $\pm 0.2 Pa$ | $0 - 60 Pa$ |
| 3-**hole probes:** | | |
| 1.- Sensortechnics $HCL0075E$ | $\pm 3.25 Pa$ | $\pm 7500 Pa$ |
| 2.- Phase standard deviation | $1 - 3 Pa$ | $50 - 210 Pa$ |
| 3.- Angle of attack, $\alpha$ | $0.3°$ to $1.2°$ | $0°$ to $10°$ |
| **Pressure taps:** | | |
| 1.- Sensortechnics $HCL0025E$ | $\pm 1.25 Pa$ | $\pm 2500 Pa$ |
| 2.- Phase standard deviation | $2 - 4 Pa$ | $40 - 300 Pa$ |
| 3.- Angle of attack, $\alpha$ | $0.2°$ to $1.3°$ | $-2°$ to $11°$ |

During the measurement campaign, while the changes on the pitch or yaw angle were made between test cases, the tunnel was left open to allow for fresh air to enter the tunnel circuit. As a result, the temperature and relative humidity were kept within $18 \pm 1.5°$C and $40 \pm 5\%$, respectively. According to Tsilingiris (2008), these values represent small changes in the physical properties, thus, a density correction was neglected.

 **4 Results and discussion**

The results are presented in this section, starting from the pressure distributions and the relative dynamic pressure along the chord at the span position of $r = 45\%R$, followed by the comparison between the described methods to determine the AoA. Finally, an additional comparison is presented with the variations of the pitch angle.

## 4.1 Pressure distribution

The AoA estimation based on the surface pressure measurements depends on the relative dynamic pressure ($q_{rel}$) and the pressure difference ($\Delta P(12.5\%c)$), see Eq. 6. It is hence important to examine their variation with azimuth position before proceed to the AoA estimation. Figure 11 shows both the variation of both variables normalized by the freestream dynamic pressure $q_\infty = 0.5\rho U_\infty^2 \approx 25Pa$.

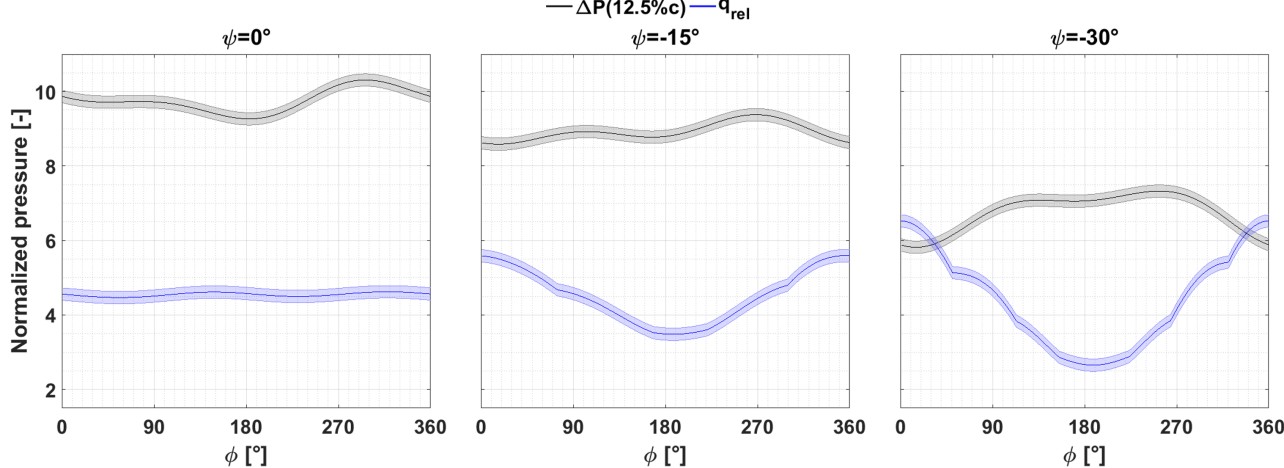

**Figure 11.** Results from pressure taps at $r = 45\%R$. For three yaw angles, relative dynamic pressure ($q_{rel}$) and pressure difference between the pressure and the suction side of the blade at $12.5\%c$ variations with azimuth angle. Values are normalized by the dynamic pressure $q_\infty$

For the aligned case, $\psi = 0°$, the relative dynamic pressure remains relatively constant at $q_{rel} = 4.5\ q_\infty$, while the pressure
difference at $12.5\%c$ exhibits four marked behaviours:

Initially, $0° \leq \phi \leq 90°$, it remains relatively constant at $\Delta P(12.5\%c) = 9.8\ q_\infty$. Then the dynamic pressure drops, to reach a minimum at $\phi = 180°$ ($9.3\ q_\infty$), while an increase follows from $\phi = 180°$ to $\phi = 290°$. At that point, the dynamic pressure reaches its maximum value ($10.3\ q_\infty$) before it starts dropping to reach $9.8\ q_\infty$ at $\phi = 360°$.

This behavior agrees qualitatively with computational results made by Schulz et al. (2017), where it is shown an asymmet-
340 rical axial load, even without the presence of yaw misalignment.

With the introduction of yaw misalignment $\psi = -15°$, the relative dynamic pressure is influenced by the yaw angle, showing a symmetrical trend with its minimum value at an azimuth angle of $\phi = 180°$. The maximum variation is $\Delta q_{rel} = q_{rel,max} - q_{rel,min} = 2\ q_\infty$. The pressure difference at $12.5\%c$ displays similar features as in the aligned case, but with a shifted azimuth

angle position, getting its minimum, $\Delta P(12.5\%c) = 8.5\,q_\infty$, at $\phi = 0°$ and it maximum, $\Delta P(12.5\%c) = 9.5\,q_\infty$, at $\phi = 270°$.
This behavior suggests being related to the advancing/retreating behavior described by Schulz et al. (2017).

For the case of yaw angle $\psi = -30°$, the relative dynamic pressure behavior remains and the drop increases up to $\Delta q_{rel} \approx 3.8\,q_\infty$. In the case of the pressure difference at $12.5\%c$ the azimuth angle dependency becomes more important and the advancing/retreating influence is more pronounced, producing a plateau between azimuth angles $90° \leq \phi \leq 270°$

In terms of the measurement range, the relative pressure is $2.8 \leq q_{rel}/q_\infty \leq 6.5$. Over this range, the uncertainty error
represents the $4.5\%$. In the case of the pressure difference at $12.5\%c$, the range is $6 \leq \Delta P(12.5\%c)/q_\infty \leq 10.3$, where the error takes a value of $4\%$.

The magnitude of the dynamic pressure, $q_{rel}$, and the location of the stagnation point fluctuate along the azimuth position in the misaligned cases. Figure 12 provides an overview of the stagnation point location and the pressure magnitude variation for the different yaw cases in the region close to the leading edge ($0\%c \leq x \leq 4\%c$). The position of the stagnation point at each azimuth angle is indicated on the pressure contours by circles ($\circ$).

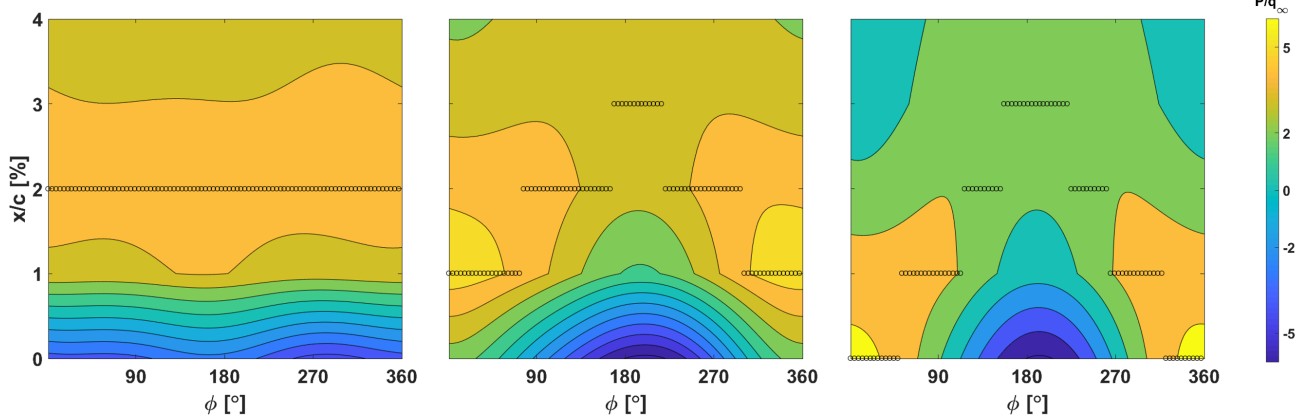

**Figure 12.** Pressure contours over the pressure side at $r = 45\%R$ in the range $[0,4]\%c$ for all yaw cases and pitch angle of $\theta = 0°$. The circles ($\circ$) are located at $max\{P\}$ at that azimuth position and indicate the location of the stagnation point.

It can be seen that for yaw angle $\psi = 0°$ case, Fig. 12 (left), the relative dynamic pressure position is always at $x = 2\%c$. On the contrary, for the yaw angle $\psi = -15°$ case, Fig. 12 (middle), the stagnation point is farther upstream ($x = 1\%$) at azimuth angle $\phi = 0°$ and moves downstream towards $x = 3\%$ for $\phi = 180°$, and back to $x = 1\%$ as the blade moves towards the $\phi = 0°$ position. Finally, for the case of yaw $\psi = -30°$, Fig. 12 (right), the behavior of the stagnation point is similar, but
more pronounced, between at $x = 0\%$ and $x = 3\%$ at azimuth angles of $\phi = 0°$ and $\phi = 180°$, respectively.

The pressure taps are located at discrete points on the blade surface. For this reason, the sensor that estimates the stagnation point, i.e. the values of the relative dynamic pressure, fluctuate in location. The latter explains the sharp changes present in yaw angle $\psi = -15°$ at azimuth angles $\phi \approx 70°$ and $\phi \approx 300°$ and yaw angle $\psi = -30°$ at azimuth angles of $\phi \approx 50°$ and $\phi \approx 320°$(see Fig. 11).

Regarding the drop in relative dynamic pressure for the misalignment cases, this can be explained with the geometrical velocities. Equation 10 shows both, normal and tangential contributions, resulting from the relative dynamic pressure $q_{rel,geo} = 0.5\rho U_{rel}^2$ (see Eqs. 7 and 8).

$$\frac{q_{rel,geo}}{q_\infty} = \underbrace{(cos(\psi))^2}_{normal\ contribution} + \underbrace{(\lambda(r/R) - sin(\psi)cos(\phi))^2}_{tangential\ contribution} \tag{10}$$

    Figure 13 shows the relative dynamic pressure at the radial position $r = 45\%R$ for the aligned and misalignments cases,
normalized by the freestream dynamic pressure $q_\infty$. It can be seen the same trend between the geometrical case (dashed line) and the estimation from the pressure taps (PP, solid line) as well in the maximum ($\phi = 0°$) and minimum ($\phi \approx 180°$) azimuth positions.

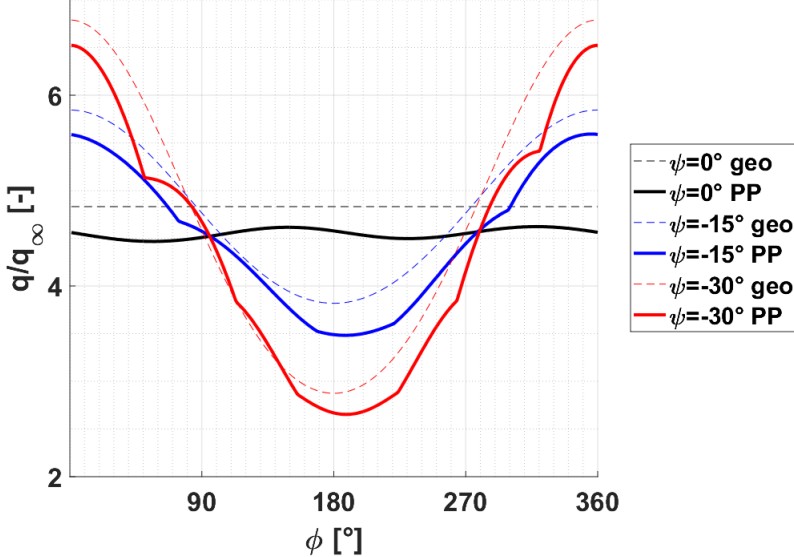

**Figure 13.** Normalized relative dynamic pressure at radial position $r = 45\%R$ for the yaw cases. Solid line, pressure tap estimation. Dashed line, geometrical calculation.

## 4.2 Angle of attack estimation

### 4.2.1 Test cases

Figures 14, 15 and 16 show the AoAs results from the pressure tap ($PP$ 45%$R$) and the 3-hole probe ($3HP$) methods over the three yaw angles cases. In the interest of clarity, only one of the pitch angles is presented here for each yaw angle case. For completeness, the results for the remaining pitch cases can be found in App. E and an analysis through the pitch cases is presented in Sect. 4.2.2.

Figure 14 shows the AoA for the pressure taps and 3-hole probes approaches (left) and the analytical calculations (right) at 380  pitch angle $\theta = 0°$ in the aligned case. It can be seen that the two approaches are able to capture the tower influence, which produces a reduction of the AoA around the azimuth angle of $\phi = 180°$. However, the AoA from the 3-hole probes method capture a drop near the zone of azimuth angles $\phi \approx 90°$ and $\phi \approx 290°$. This behavior has been seen in previous results of Klein et al. (2018); Bartholomay et al. (2018); Marten et al. (2018).

The explanation is due to the heterogeneity of the inflow. These variations, $\Delta U_\infty = \pm 0.2 ms^{-1}$ (see Fig. 2), can have the 385  same influence as the tower over the AoA estimations. The geometrical estimation ($\alpha_{geo}$) under such inflow variations, results in an AoA difference of $\Delta \alpha_{geo} = \pm 0.4°$, which supports this statement.

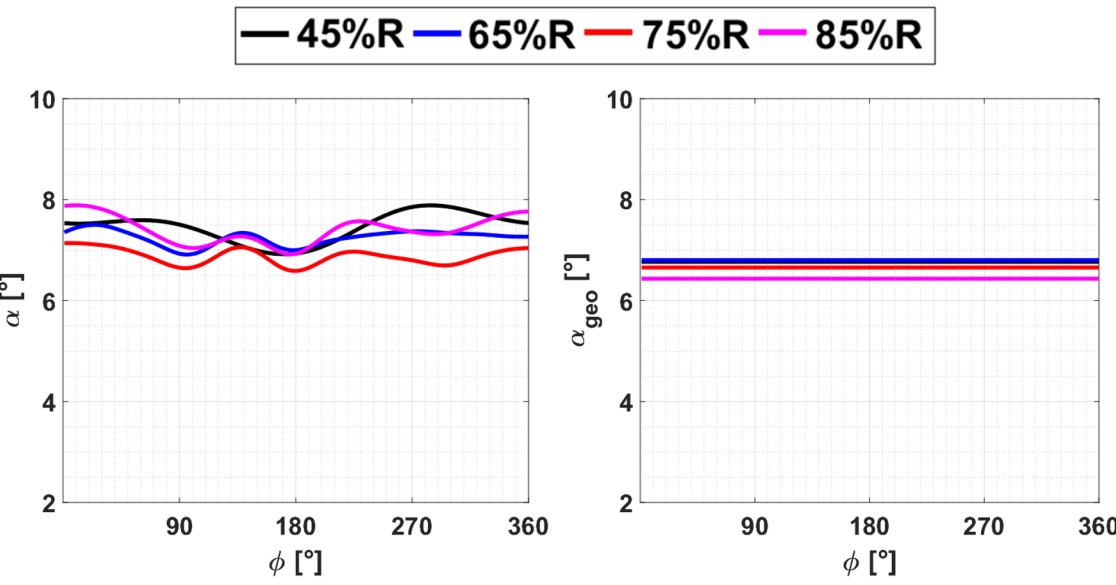

**Figure 14.** AoA results for yaw angle $\psi = 0°$ and pitch angle $\theta = 0°$. Pressure taps and 3-hole probe approaches (left). Analytical calculations (right).

Although the AoA over the azimuthal variation is not constant, both methods estimate a similar AoA range. The AoA for both pressure tap and 3-hole probe methods are slightly lower than previous experimental results show by Klein et al. (2018),

but within the uncertainty values. Table 3 shows the range ($\alpha_{min}, \alpha_{max}$) and average ($\overline{\alpha}$) values of the AoA over the azimuth
angle for the pressure taps and the 3-hole probe methods. The range of the tools measurements is between $6.6 - 7.8°$ and the
geometrical estimation between $6.4 - 6.8°$.

On previous work by Klein et al. (2018); Marten et al. (2018) the AoA estimations made with far field considerations showed
an offset of $\Delta\alpha_{off} = 2.3°$ respect to the 3-hole probes. The smaller difference between experimental and analytical estimations
in the current work supports the fact that the blockage model is well implemented.

Additionally, Table 3 shows a comparison between the pressure tap and each 3-hole probe. The overall average AoA differ-
ence, $\overline{\Delta\alpha} = mean\{|\alpha_{PP} - \alpha_{3HP}|\}$, shows that there is a small difference between the pressure tap and 3-hole probe methods,
up to $\overline{\Delta\alpha} = 0.6°$. Whereas the AoA maximum difference, $\Delta\alpha_{max} = max\{|\alpha_{PP} - \alpha_{3HP}|\}$, located around the azimuth angle
of $\phi \approx 300°$ takes the values of $\Delta\alpha_{max} = 1.2°$. However, the difference is in the same magnitude that of the fluctuations of
each tool.

**Table 3.** AoA from the pressure taps and 3-hole probe methods at yaw angle $\psi = 0°$. Average, minimum and maximum for the pitch angle
case $\theta = 0°$.

| Method | $\overline{\alpha}$ [°] | $\alpha_{min}$ [°] | $\alpha_{max}$ [°] | PP comparison | | |
|---|---|---|---|---|---|---|
| $PP$ 45%R | 7.4 | 6.9 | 7.8 | $\Delta\alpha_{max}$ [°] | $\overline{\Delta\alpha}$ [°] | $std(\Delta\alpha)$ [°] |
| $3HP$ 65%R | 7.2 | 6.9 | 7.5 | 0.6 | 0.3 | 0.2 |
| $3HP$ 75%R | 6.8 | 6.6 | 7.1 | 1.2 | 0.6 | 0.3 |
| $3HP$ 85%R | 7.3 | 6.9 | 7.8 | 0.6 | 0.2 | 0.2 |

Figure 15 shows the AoA from the pressure tap and 3-hole probe methods (left), and analytical calculations (right) for the
pitch angle $\theta = 0°$ and the yaw misalignment of $\psi = -15°$.

From Fig. 15 (left), it can be noticed that the AoA estimation from the pressure tap starts with smaller values until azimuth
angle $\phi \approx 90 \pm 20°$ where becomes larger than the AoA from the 3-hole probes estimation. The 3-hole probe approach still
shows the tower influence with a drop in the AoA around the azimuth angle $\phi = 180°$, in contrast with the pressure tap method,
where the AoA keeps increasing until the maximum position located in azimuth angle of $\phi \approx 200°$. A reduction in the AoA is
followed where the pressure tap estimation becomes smaller than the 3-hole probe approach, as the blade is moving towards
the azimuth angle $\phi = 0°$.

The same behavior is presented in the case of analytical AoA, Fig. 15 (right) with two main differences, first, there is no tower
effect, due to the analytical approach is not taking this into consideration. Second, a particular behavior is noticed regarding
the 3-hole probes at $75\%$ and $85\%R$, where their positions are shifted. This could be cause by an error in the mounting, due to
it is visible also without misalignment (Fig. 14).

For this yaw misalignment, it is shown that the 3-hole probe has a trend less pronounced than the pressure tap approach
between $0° \leq \phi \leq 90°$ and $270° \leq \phi \leq 360°$. Furthermore, the crossflow has covered partially the influence of the tower in the
pressure tap method, increasing the AoA disagreement between both methods is in the azimuth angle range $135° \leq \phi \leq 225°$.

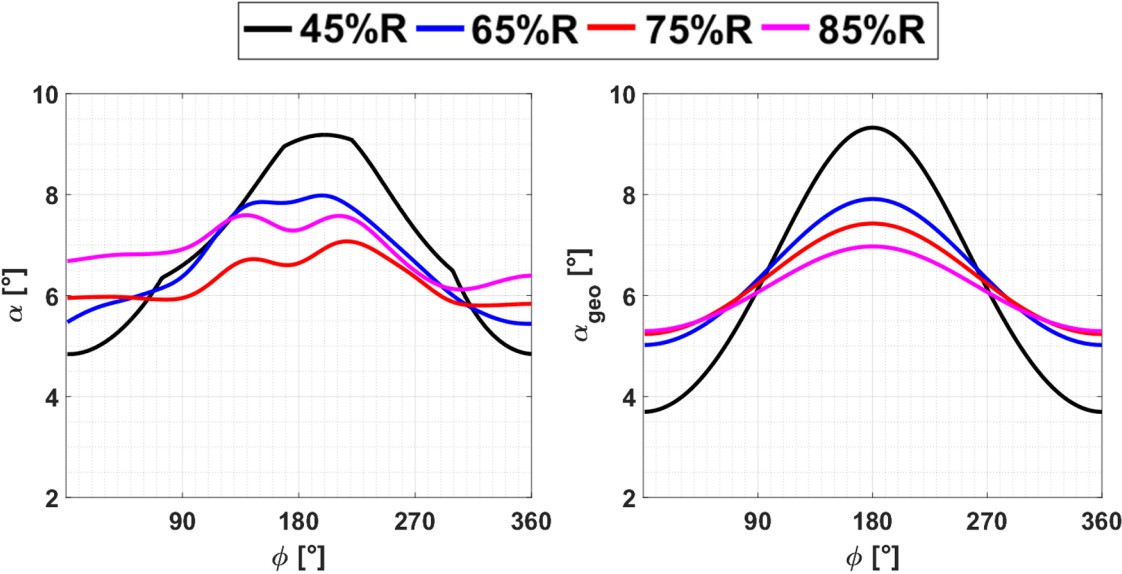

**Figure 15.** AoA results for yaw angle $\psi = -15°$ and pitch angle $\theta = 0°$. Pressure taps and 3-hole probe approaches (left). Analytical calculations (right).

Figure 16 shows the AoA from the pressure tap and 3-hole probe methods (left), and analytical calculations (right) for the pitch angle $\theta = 0°$ and the yaw misalignment of $\psi = -30°$.

The behavior of the AoA results from the pressure tap method, Fig. 16 (left), in this case, is similar to the yaw angle $\psi = -15°$, exhibiting a more pronounced difference with the 3-hole probe approach in the azimuth angle $\phi = 180°$. The effect of the crossflow due to the yaw misalignment is dominant in this case, diminishing the AoA drop around the azimuth angle $\phi = 180°$ in the 3-hole probe and with a steeper maximum in the case of the pressure tap, in contrast with the previous yaw case.

The analytical AoAs, Fig. 16 (right), show the same features, including the large difference at azimuth angles $\phi = 0°$ and $\phi = 180°$.

Overall, the pressure tap method presents good results, qualitatively and quantitatively. In the aligned case, the average difference between 3-hole probes and analytical AoA is below $1°$. Under yaw misalignments, the pressure tap method in comparison with the analytical method shows an average difference of $\overline{\Delta\alpha} = 0.8$ and $\overline{\Delta\alpha} = 1.2$ for yaw angles $\psi = -15°$ and $\psi = -30°$, respectively. The larger differences are presented in azimuth angle $\phi = 0°$.

### 4.2.2 Pitch analysis

A comparison between the AoA estimations from both approaches trough the pitch angle cases, in a fixed azimuth position, $\phi = 315°$, was analyzed. Figure 17 shows the evolution of AoA estimations at the azimuth angle of $\phi = 315°$. It can be observed

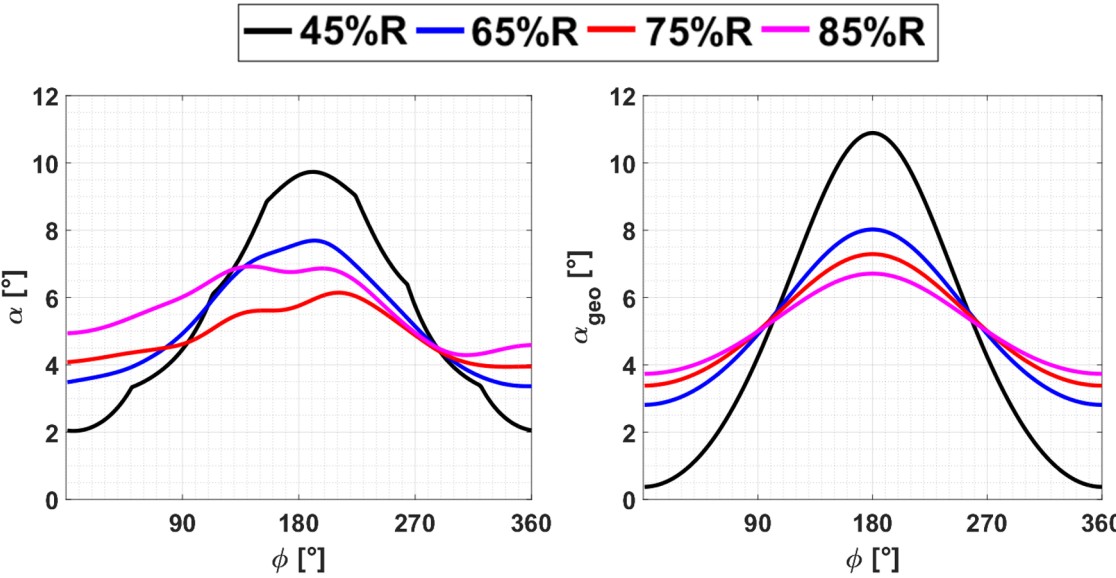

**Figure 16.** AoA results for yaw angle $\psi = -30°$ and pitch angle $\theta = 0°$. Pressure taps and 3-hole probe approaches (left). Analytical calculations (right).

that the trend is linear for both methods. While the yaw angle increases the pressure tap method change from estimate larger to smaller values than 3-hole probes.

A linear fit $\alpha = m\theta + k$ was obtained, in order to check the relation between AoA and pitch angle. The slopes take values around $m = -0.7 \pm 0.1 [1/°]$. From the geometrical point of view (see Eq. 9), the expected slope between the AoA and pitch is $m = -1$. Nevertheless, the induction factors change at each pitch angle, therefore the change in the slope is the results of that dependency. This agrees with the fact that the slopes are similar but not the same, as is expected variations of the induction factor along the radial positions.

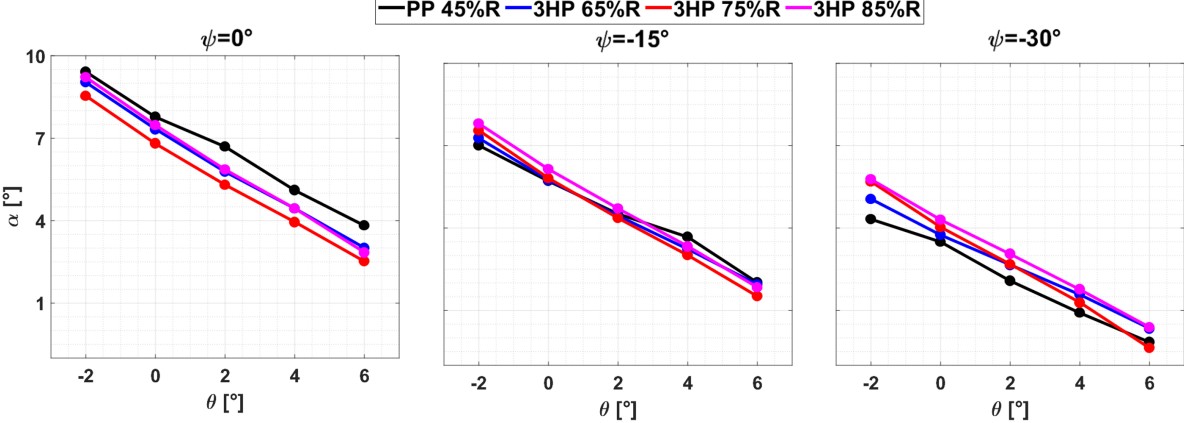

**Figure 17.** AoA estimations from pressure tap and 3-hole probe methods, variations with pitch angle. Three yaw cases $\psi = 0, -15, -30°$.

## 5    Conclusions

A method to determine the AoA based on the pressure difference between the pressure and suction side on a wind turbine blade
was tested. The method was compared with the AoA results from three 3-hole probes in simultaneous wind tunnel measurements together with analytical calculations. Several conditions were studied regarding the introduction of yaw misalignment
and different pitch angles for the blades.

The pressure distribution on the blade at $45\%R$ was measured through chordwise pressure taps. The tested method uses
the information of a reduced number of pressure taps located close to the blade leading edge in order to estimate the relative
dynamic pressure to its corresponding blade section. Additionally, the pressure difference between suction and pressure side
of the blade at $12.5\%c$ is tracked in order to determine the AoA based on 2-D assumptions.

The application of the method can be summarized as follows:

1. 2D calculations:

   (a) Perform computational calculations or 2D airfoil measurements to obtain the pressure distribution $C_P$ of the same
       profile to study on 3D.

   (b) Get a fit equation between the pressure difference of lower and upper side $\Delta C_P$ at $12.5\%c$ and AoA:
       $\Delta C_P(12.5\%c) = k_1\alpha + k2$

2. 3D estimations:

   (a) Perform pressure distribution measurements at a blade section with similar characteristics of the 2D airfoil. Only
       pressure taps at $12.5\%c$ are needed.

   (b) Identify the relative dynamic pressure, $q_{rel}$, at the azimuth station. The method of the stagnation point was here
       presented. Pressure taps at the leading edge vicinity would be needed.

(c) Estimate the AoA through the inverse equation from the 2D calculations: $\alpha = \dfrac{1}{k_1}\left(\dfrac{\Delta P(12.5\%c)}{q_{rel}} - k_2\right)$

The main restrictions are the use of an thin airfoil and attached flow.

The results show that in the aligned case, $\psi = 0°$, the pressure tap approach is suitable, being capable of capturing the same features of the AoA results from the 3-hole probes, including the influence of the tower effect. The comparison between the pressure tap method and the three 3-hole probes present a maximum average difference of $\overline{\Delta\alpha} = 0.6$.

With the introduction of yaw misalignment, the AoA results from the pressure tap method show, as expected, the crossflow influence in a more pronounced curve than the 3-hole probe, in agreement with the analytical results. The crossflow impact
is more dominant than the tower effects and the pressure tap method is not able to predict its influence, from where it can be inferred an AoA overestimation in the azimuth region of $135° \leq \phi \leq 225°$.

Regarding the pitch angle changes in the blades, the AoA results from the pressure tap approach presents a linear behavior with a slope value of $|m| \approx 0.7[1/°]$, similarly to the 3-hole probe method, being capable to capture the resulting effects from the axial and tangential induction.

Overall, it is found that the pressure tap method applied here to determine the AoA, provides reliable data, with good performance for both aligned and misaligned cases. Hence, the presented method is a promising alternative to the use of external probes, which affect the flow over the blade and require additional calibration.

*Data availability.* Pressure measurement data and results can be provided by contacting the corresponding author

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

# Appendix A:  List of symbols

| | |
|---|---|
| $\alpha$ | Angle of attack |
| $U$ | Velocity |
| $\psi$ | Yaw angle |
| $\phi$ | Azimuth angle |
| $\lambda$ | Tip speed ratio |
| $f$ | Rated frequency |
| $R$ | Rotor radius |
| $\gamma$ | Twist angle |
| $\theta$ | Pitch angle |
| $c$ | Chord length |
| $r/R$ | Nondimensional radial blade position [0,1] |
| $x$ | Horizontal chord position |
| $\mathbf{x}$ | Nondimensional chordwise coordinate [0,1] |
| $y$ | Vertical chord position |
| $X$ | Axial wind tunnel position |
| $Y$ | Lateral wind tunnel position |
| $Z$ | Vertical wind tunnel position |
| $R^2$ | Coefficient of determination |
| $\rho$ | Air density |
| $\Omega$ | Angular velocity |
| $q$ | Dynamic pressure |
| $g$ | Gaunaa model contribution in pressure distribution |
| $\beta$ | Flap angle |
| $k$ | Fit constant |

## Appendix B:  Abbreviations

| | |
|---|---|
| $PP$ | Pressure tap method |
| $3HP$ | 3-hole probe method |
| $BeRT$ | Berlin Research Turbine |
| $AoA$ | Angle of attack |

## Appendix C:  Subscripts

| | |
|---|---|
| $\infty$ | Free stream |
| $ref$ | Reference value |
| $upper$ | Blade section suction side |
| $lower$ | Blade section pressure side |
| $s$ | sensor |
| $corr$ | Corrected value |
| $probe$ | In reference of probe coordinate system |
| $probe, section$ | In reference of blade section coordinate system |
| $rel$ | Relative |
| $c$ | Circulatory |
| $eff$ | Effective |
| $camb$ | Camber |
| $L$ | Nonlinear terms |
| $t$ | Tangential |
| $n$ | Normal |

## Appendix D: Uncertainty of the angles of attack

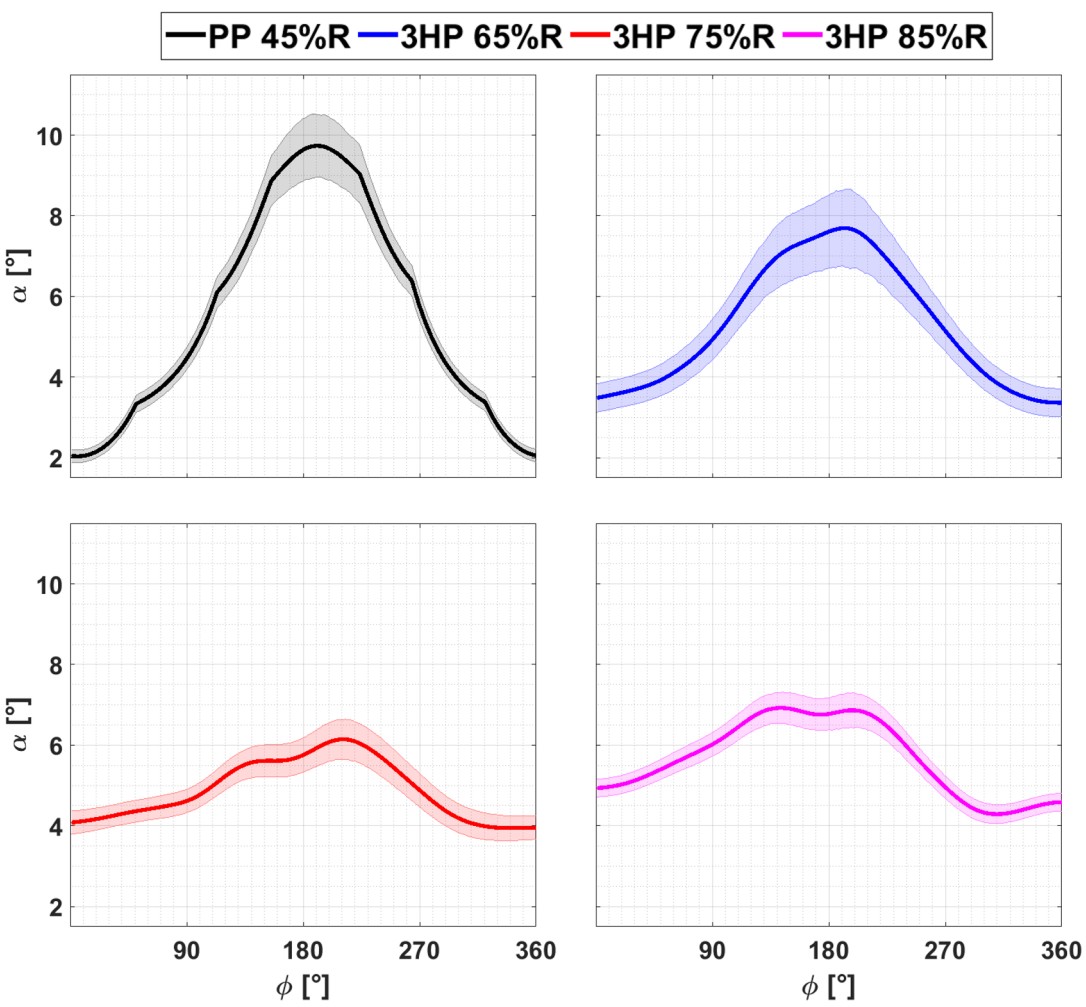

**Figure D1.** AoA results from the pressure tap and 3-hole probe approaches with their uncertainties. Pitch angles: $\theta = 0°$ and yaw angle $\psi = -30°$.

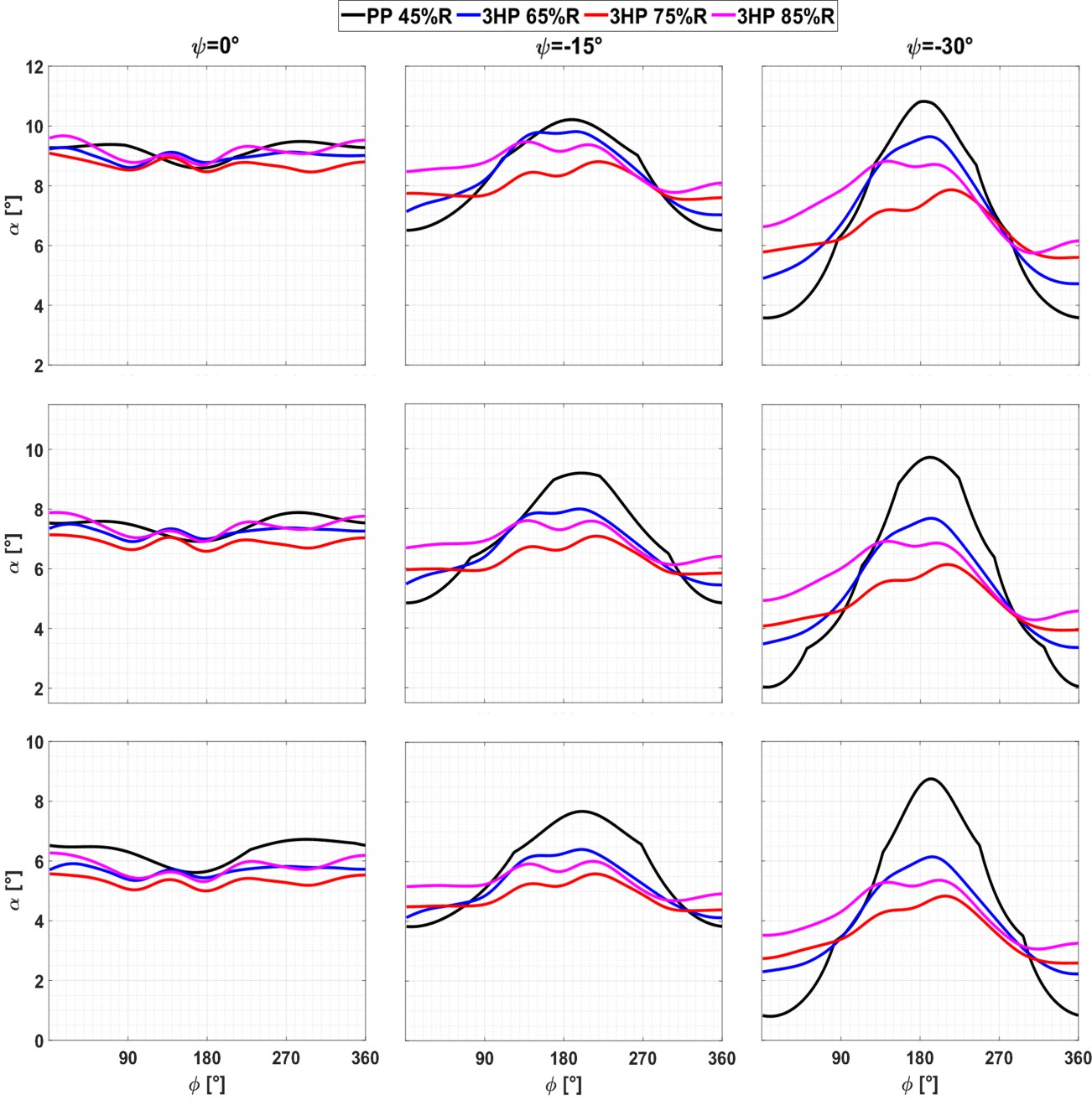

**Figure E1.** AoA results from the pressure tap and 3-hole probe approaches. In columns yaw angles: $\psi = 0°$, $\psi = -15°$ and $\psi = -30°$. In rows pitch angles: $\theta = -2°$, $\theta = 0°$ and $\theta = 2°$.

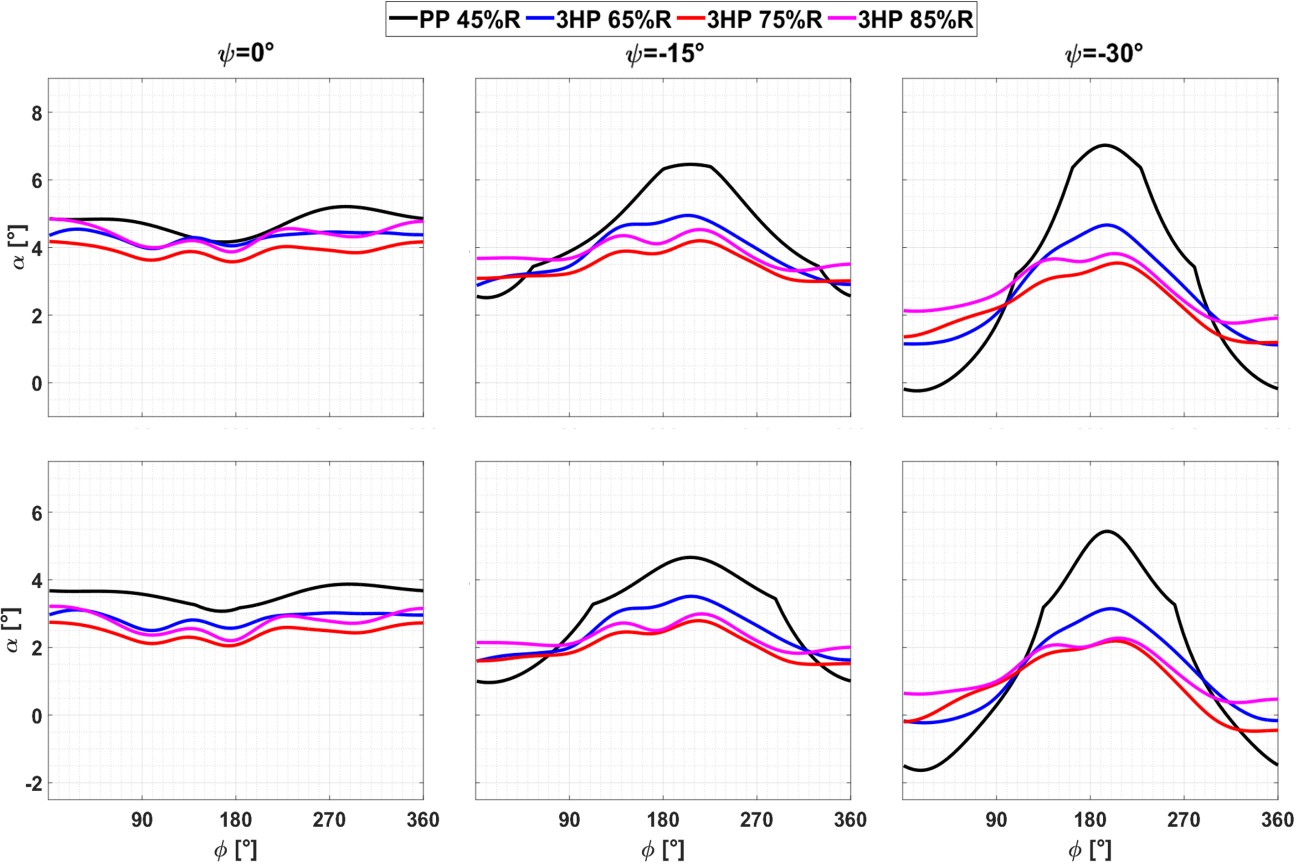

**Figure E2.** AoA results from the pressure tap and 3-hole probe approaches. In columns yaw angles: $\psi = 0°$, $\psi = -15°$ and $\psi = -30°$. In rows pitch angles: $\theta = 4°$ and $\theta = 6°$.

*Author contributions.* RSV carried out the measurement campaign with the support of JA and SB. RSV worked in the implementation of the pressure tap method, performed the calculations, analysis and wrote the paper. SB provided the code for the 3 -hole probe method. JA, SB, MM, CNN, and COP contributed with comments and discussions about each section in the manuscript.

*Competing interests.* The authors declare that they have no conflict of interest.

*Acknowledgements.* R. Soto-Valle would like to thank the support of ANID PFCHA/Becas Chile-DAAD/$2016 - 91645539$. The authors would also like to acknowledge J. Saverin for providing valuable feedback.

585