# Peer review of "Determination of the Angle of Attack on a Research Wind Turbine Rotor Blade Using Surface Pressure Measurements"

_Wind Energy Science, 2020_

## Referee Comment (RC1) · Uwe Paulsen (Referee) · 27 Apr 2020

Determination of the Angle of Attack on a ResearchWind Turbine

overall:

this manuscript is not easy to read and would need a structuring in the chapters, due to lack of missing overviews of the setup and issues from the previous experiment which is not mentioned appropriately.(by accessing the references i can get the picture)

The authors have not performed thorough litterature review covering the aspects from testing research turbine/wind tunnel test on models.

Also scaling is not mentioned as a source for uncertainty in experimental tests.

The neglect of measuring sideslip in velocity measurements may be intensional or not.This is not clearly communicated, nor documented.

The damping characteristics of the used accustic tubing setup is not appropriately documented and the results bear footprints of undocumented or missing research

-------------------------------------------------------o-------------------------------------------------------------------------
* * *
Rotor Blade Using Surface Pressure Measurements

Abstract: may be rephrased du to conclusions, based on edititorial issues.

Introdction:

L16: please make it more clear why is AoA a challenge?There are practical solutions to measure inflow, but is it aoa? The sentence/question should be open up for some refections essential for the motivation.

It would be great for the reader to have a (tablar)listing of available methods.

L31 At Risø(DTU) inflow measurements on a real turbine was conducted already in 90ties  by Risø Nat. labs, and NREL around same time

L46 it depends on the blade length and scaling; a 20mm pitot tube in comparison with a 60m blade-please adjust this against windtunnel testing

L48 no references given

L51 unclear sentence: with a pitot pressure sensors you know the position geometrically.

2: Experimental setup

Figure 2 is missing definitons ($U_t$, $U_n$, $U_{rel}$), as well as t/c

L109: explain 'on a comparable level', e.g what is the implicit effect of Turbulence(1.5%) ??

The choice for using  ClarkY is not clear(high drag airfoil), see f.example DOI: 10.2514/6.2006-33 Conference: 44th AIAA Aerospace Sciences Meeting and Exhibit

L113: Model Blockage and conseqences for interpretation of results?

L115: is the turbine yaw fixed or free?

L124: the statement of placement of pressure taps is not constant=0.45-why straight line placement?/why is it in this small scale experiement not following constant radius?

L126: what is the max frequency(3 dB limit) of the detectable signal

L129:specs?

L134: A miss why the use of flaps with consensus on title /intro & science objectives

L157: using a 3 hole probe-no side slip detection. What about the flow conditions when the turbine is in yaw?

L171: what is the explanation behind seeing the 1P in the signal for the interpretation?

L174: This is a surprising statement about resonator box system that doesn't damp frequencies.30 Hz filter? The cited reference(Berg) offers fig 21(assuming small tubes) with considerable amplitude and phase lag properties.This needs clarification

Fig 8: What is the difference between the black-dashed and red pointsaroud x/c=0.3..and onwards?

L237: the discussion of cross flow(sideslip) for the 2D probe is missing.Or may be your statemnt is to use a 2D probe in the 3D inflow as a representation of the normal(tangential) velocity components? Clarification and error calculation is needed..

L253: temperature increases in the flow during experiments effects on the pressure sensors(standard calibration at 25deg nom)? As I recall the HCL's have +-0.25%FS nonlinearity & hysteresis. So i would assume higher uncertainty on aoa ..

Table1 needs to state that uncertainty is [fraction/%] of FSR

4.1 Pressure distribution

the question is if yaw affetcs the pressure in the dynamic inflow field, observed here with a 2D-probe.

The results are expressed in Pascal, may be it is more clear to show it relativel(normalisation), speaking of uncertainty and also from a point of measurement range.

Figure 11: add of result for 0 yaw missing

Figure12: Odd..with the 2-2½P variations(L316), except for the tower influence... Check!

L334: Could this be the damping effects from the resonating tubes characteristic ?, same P VARIATION ISSUE AS ABOVE

conclusion: may change in details.

---

## Referee Comment (RC2) · Anonymous Referee #2 · 5 May 2020

General comments: Interesting paper that shows the potential for an AoA estimation from a measured surface pressure in a few chord wise locations.

The paper presents some interesting measurements on a scaled rotor and it would be obvious to try the method on a full-scale rotor.

Overall, the method and results are presented clearly and thoroughly. However, before the paper can be published the wording of some of the paragraphs should be rephrased, as it is sometimes difficult to understand the argumentations.

Specific comments:

[Figure]

Line No Comment

139 Is there a reason for this very high logging frequency?

148 Check how the references are referenced

187 Can this fit be shown and has the fit been checked with measurements from the blade at standstill? If this is possible?

200 Can you elaborate a little more on this? Why is it neglected and what (if any) consequences does it have?

227 It would be good with a comment on how good the fit is or perhaps a figure that shows the fit.

231 Why did you choose this position and not one where there were pressure taps, so you can avoid the interpolation?

247 Change "Equation 7 it can..." to "Equation 7 can..."

Table 1 What is "FSR"?

274 Rephrase the sentence, especially the part with "...to after drops..."

297 Rephrase the sentence. It is an important observation, but difficult to understand as it is written here.

303 & List of symbols Missing expression for the tip speed ratio.

Figure 12 Legend is different from Figures 13, 14 and 15.

Figure 12 Perhaps repeat that AoAgeo = 5.1 deg from Line 249.

321 Can the uncertainties be added on the figure?

322-23 Rephrase

324-26 Can the effect of the walls be quantified, e.g. by the method from H. Glauert

The elements of aerofoil and airscrew theory?

324-26 Rephrase

330-31 Rephrase

353-56 The paragraph is unclear, please rephrase

363 Is there an explanation on the difference in slope? Why is the measurements not showing a slope of -1?

364-66 Rephrase
* * *

---

## Author Comment (AC1) · 12 May 2020

**Determination of the Angle of Attack on a ResearchWind Turbine Rotor Blade Using Surface Pressure Measurements.**
Reply to Uwe Paulsen (Referee)

Dear Dr. Paulsen (Referee), below you can find the answers to your comments

**Abstract:** May be rephrased due to conclusions, based on edititorial issues.
- Reply: It will be rewritten based on the applied changes.

**1 Introduction:**
L16: please make it more clear why is AoA a challenge? There are practical solutions to measure inflow, but is it aoa? The sentence/question should be open up for some reflections essential for the motivation. It would be great for the reader to have a (tablar) listing of available methods.
- Reply: This will be implemented.

L31 At Risø(DTU) inflow measurements on a real turbine was conducted already in 90ties by Risø Nat. labs, and NREL around same time.
- Reply: These references will be included as part of the introduction.

L46 it depends on the blade length and scaling; a 20mm pitot tube in comparison with a 60m blade-please adjust this against wind tunnel testing.
- Reply: Agreed, this will be rephrased including the scaling reference mentioned.

L48 no references given.
- Reply: will be included.

L51 unclear sentence: with a pitot pressure sensors you know the position geometrically.
- Reply: Agreed, the sentence wants to point out that the pressure taps and probe are not exactly in the same radial position, therefore the AoA provided by the probe is not the AoA at the pressure taps. This sentence will be rephrased.

**2 Experimental setup**
Figure 2 is missing definitons (Ut, Un, Urel), as well as t/c.
- Reply: This will be included.

L109: explain 'on a comparable level', e.g what is the implicit effect of Turbulence (1.5%)?
- Reply: An additional explanation will be included in order to fullfill the turbulence calculation and its repercussions.

The choice for using ClarkY is not clear(high drag airfoil), see f.example DOI: 10.2514/6.2006-33 Conference: 44th AIAA Aerospace Sciences Meeting and Exhibit
- Reply: Additional information about the used airfoil will be provided. Regarding the high values in the Cl/Cd plot are for small Reynolds number (<10^5). The latter is not the case when the turbine is set at rated condition (Reynolds number based on the blade chord and relative velocity is in the range of 1.7-3.0 x 10^5)

L113: Model Blockage and consequences for interpretation of results?
- Reply: The blockage effect has been modelled computationally and it was found sinificant in terms of power and thrust, compared to field conditions. The latter,

it motivated the pitch steps, checking the sensibility of the estimations approaches, supporting that the pressure taps method will be able to capture a realistic expansion in the absence of walls. The blockage effect will be discussed in the revised submission.

L115: is the turbine yaw fixed or free?

- Reply: Is fixed and the change can be done only manually before any measurements. This will be explicit in the final document.

L124: the statement of placement of pressure taps is not constant=0.45-why straight line placement?/why is it in this small scale experiement not following constant radius?

- Reply: This is correct. They are in a straight line perpendicular to the spanwise line from the root until 45%R at x/c=0.3. The curvature error is considered small ($\Delta r < 25mm$ ). However, it was considered when the pressure was corrected by centrifugal effect as shown L165. This will be explicit.

L126: what is the max frequency (3 dB limit) of the detectable signal

- Reply: This will be included.

L129:specs?

- Reply: The spectra from the three-hole probes will be included.

L134: A miss why the use of flaps with consensus on title /intro & science objectives

- Reply: The TE-flaps were set in their neutral position for all the experiments, and they are exposed only for completeness of the equipped blade information.

**3 Methodology**

L157: using a 3 hole probe-no side slip detection. What about the flow conditions when the turbine is in yaw?

- Reply: In the case of the cross-flow presence, the calibration loses its one-to-one relationship with a probe in a yaw angle (becoming multi-valued). This is the case of a combination of a large angle of attack and yaw angle or an excesive yaw angle. For the study cases, the angle of attack remains below ~11° for every azimuth, yaw, and pitch angles which suggest a small influence, this error will be included, regarding both misalignment  (Figure 4 of Pisasale, A. J., and N. A. Ahmed. "A novel method for extending the calibration range of five-hole probe for highly three-dimensional flows." Flow Measurement and Instrumentation 13.1-2 (2002): 23-30.)

L171: what is the explanation behind seeing the 1P in the signal for the interpretation?

- Reply: 1P corresponds to the tower effect.

L174: This is a surprising statement about resonator box system that doesn't damp frequencies.30 Hz filter? The cited reference(Berg) offers fig 21(assuming small tubes) with considerable amplitude and phase lag properties.This needs clarification.

- Reply: This will be discuss in detail, including the lag and amplitude ratio estimations from the reference cited.

Fig 8: What is the difference between the black-dashed and red points around x/c=0.3..and onwards?

- Reply: The difference will be discussed in the next version.

L237: the discussion of cross flow(sideslip) for the 2D probe is missing. Or may be your statemnt is to use a 2D probe in the 3D inflow as a representation of the normal (tangential) velocity components? Clarification and error calculation is needed.

- Reply: This will be discussed and supporting references will be provided, such as the one in the previous reply of side slip detection from L157.

L253: temperature increases in the flow during experiments effects on the pressure sensors (standard calibration at 25deg nom)? As I recall the HCL's have +-0.25%FS nonlinearity & hysteresis. So i would assume higher uncertainty on aoa.

- Reply: According to the manufacturer the $\pm0.25\%FS$ is in the maximum case, the nominal value is $\pm0.05\%FS$. Nevertheless, the experiments were performed measuring the wind tunnel temperature, which resulted between 17.5-19.5°C. Part of the protocol „between cases" was to leave the accesses to the tunnel opened meanwhile the change in pitch or yaw was made. This will be explicit in the new version.

Table1 needs to state that uncertainty is [fraction/%] of FSR

- Reply: This will be implemented.

**4.1 Pressure distribution**

The question is if yaw affetcs the pressure in the dynamic inflow field, observed here with a 2D-probe.

- Reply: As it was mentioned before, this will be addressed with more detail

The results are expressed in Pascal, may be it is more clear to show it relativel (normalisation), speaking of uncertainty and also from a point of measurement range.

- Reply: This will be implemented.

Figure 11: add of result for 0 yaw missing

- Reply: This will be implemented.

**4.2 Angle of attack estimation**

Fig12:Odd.with the 2-2½P variations(L316), except for the tower influence.Check! and L334: Could this be the damping effects from the resonating tubes characteristic ?, same P variation issue as above

- Reply: This will be checked in concordance with the additional comments on pressure damping.

**5 Conclusion:** may change in details.

- Reply: Overall and specific changes will be reflected in the final conclusions.

---

## Author Comment (AC2) · 12 May 2020

**Determination of the Angle of Attack on a Research Wind Turbine Rotor Blade Using Surface Pressure Measurements**

Reply to Referee #2

Dear Referee #2, below you can find the answers to your comments

L274, 297, 324-26, 330-31, 353-56 and 364-66: Rephrase
- Reply: Each of these sentences will be rewritten in order to be fully clear.

L139 Is there a reason for this very high logging frequency?
- Reply: The reason was to use the data as raw as we could. It will be included additional information regarding the filtering and its effects.

L148 Check how the references are referenced
- Reply: Reply: Thanks, it should have been a parenthesis in the reference there, it will be fixed.

L187 Can this fit be shown and has the fit been checked with measurements from the blade at standstill? If this is possible?
- Reply: Adding a Figure with this information, will only provide a linear fit (in the range of AoA that this study focus). It will be added the direct reference (equation fit) from Klein et al where this issue is highlighted. (Equation 1 in → Klein, A. C., Bartholomay, S., Marten, D., Lutz, T., Pechlivanoglou, G., Nayeri, C. N., Paschereit, C. O., and Krämer, E.: About the suitability of different numerical methods to reproduce model wind turbine measurements in a wind tunnel with a high blockage ratio, Wind Energy Science, 3, 349–460, 2018.)

L200 Can you elaborate a little more on this? Why is it neglected and what (if any) consequences does it have?
- Reply: This will be discussed. This could imply an extra added mass over the cases on transient conditions, which are out of the scope of this study.

L227 It would be good with a comment on how good the fit is or perhaps a figure that shows the fit.
- Reply: Agreed, I think that including the coefficient of determination will make the information more complete, as the fit constants are included in L226.

L231 Why did you choose this position and not one where there were pressure taps, so you can avoid the interpolation?
- Under the criterium of applying the Gaunaa's Method, is necessary to choose this position in order to avoid added mass effects due to the pitching. This will be explicit.

L247 Change "Equation 7 it can..." to "Equation 7 can..."

- Reply: Will be fixed.

Table 1 What is "FSR"?

- Reply: FSR stands for Full Scale Range, it is also included in the list of symbols.

L303 & List of symbols Missing expression for the tip speed ratio.

- Reply: The tip speed ratio is included in the list of symbols, in the fifth position.

Figure 12 Legend is different from Figures 13, 14 and 15.

- Reply: Agreed. This will be set as a unique type of legend.

Figure 12 Perhaps repeat that AoAgeo = 5.1 deg from Line 249 and L321 Can the uncertainties be added on the figure?

- Reply: This will overload the Figures. The way that is commented through the description of each Figure, it is clear and avoids the overlapping of lines in the plot while keeping the information.

L324-26 Can the effect of the walls be quantified, e.g. by the method from H. Glauert The elements of aerofoil and airscrew theory?

- Reply: Additional information regarding the blockage effects will be provided, considering the methods mentioned.

L363 Is there an explanation on the difference in slope? Why is the measurements not showing a slope of -1?

- Reply: Taking into account the equation (7), it can be noticed that the slope should be -1, but only if the other variables are independent of pitching the blades. In this case, the change of the slope is attributed to the change in the induction factor over the specific radial position. In other words „a" and „a' " are functions of the pitch angle. This will be discussed with more detail in the final version.

---

## Author Response (AR1)

**Determination of the Angle of Attack on a Research Wind Turbine Rotor Blade Using Surface Pressure Measurements.**

We would like to sincerely thank the reviewers for their time and constructive feedback. We have modified the manuscript according to your suggestions and we believe it has been significantly improved. We hope it now meets the high standards of WES journal. Please find below our answers to each of your comments. We have also provided a document where the changes are tracked (wes-2020-35-tracked_changes.pdf) and the updated version of the manuscript (wes-2020-35-Clean.pdf) with all the changes incorporated.

The referees' questions are on black, the authors' answers on blue. Additionally, the parts of the new manuscript that alludes to the comments are provided in "blue italic" for convenience.

**Referee #1**

**L16:** Please make it more clear, why is AoA a challenge? There are practical solutions to measure inflow, but is it aoa? The sentence/question should be open up for some reflections essential for the motivation. It would be great for the reader to have a (tablar) listing of available methods.

**Authors answer:** The first paragraph has been rewritten and is given below for convenience:

*"The angle of attack (AoA) is, by definition, a 2-D concept. Nevertheless, on a wind turbine, the rotating system, i.e. a blade, is under 3-D effects such as tip and root vortices, yaw misalignment, velocity inductions, among others that render the precise determination of the AoA difficult (Shen et al., 2009). Additionally, the AoA is indirectly obtained through pressure or velocity fields, thus several uncertainties are added in its estimation. In this way, determining the local AoA on wind turbine blades remains one of the greatest aerodynamic challenges. At the same time, the determination of AoA is necessary in order to calculate the lift and drag forces over the blade, develop accurate aeroelastic models, or establish a control tool."*

Henceforth the text was restructured adding new literature regarding AoA estimations, from field and wind tunnel experiments, following the structure: estimation based on probes, computational approaches, and on pressure taps. The suggestion for a table of AoA estimation methods was included and Table 1 has been added (please see answer to the next comment, too).

**L31:** At Risø(DTU) inflow measurements on a real turbine was conducted already in 90ties by Risø Nat. labs, and NREL around same time

**Authors answer:** The reviewer correctly highlights these two important contributions. We have added the following paragraph to include these campaigns and Table 1, where estimation methods of the AoA are listed:

*"Several field measurements have been conducted using probes as one of the estimation methods for the AoA. Brand et al. (1997); Simms et al. (1999); Madsen et al. (1998); Maeda et al. (2005); Bak et al. (2011a) showed measurements results employing 5-hole probes from the Energy research Centre of Netherlands (ECN), The National Renewable Energy Laboratory (NREL), Technical University of Denmark (DTU), Mie University (Mie) and DanAero projects, respectively (see Table 1). Bruining and van Rooij (1997) used 3-hole probes in the Delft University of Technology (DUT) project. The upwash*

*correction was made based on wind tunnel measurement of static blade or airfoils representative of the studied blade section. It is remarkable that the case of the ECN exhibited better results without the upwash correction. This was assumed to be a result of the compensation effect of the downwash from the shed vorticity due to the variation of the bound circulation along the blade span (Schepers et al., 2002)."*

**Table 1.** Angle of attack estimation methods on wind turbine rotor blades.

| Contributor | Blades | Radius [m] | $Re_c^a$ | On-blade tool | Estimation method |
|---|---|---|---|---|---|
| **Field** | | | | | |
| ECN[b], Brand et al. (1997) | 2 | 13.72 | 1.8M[c] | 5-hole probe, pressure taps | stagnation point, probe measurements, inverse BEM |
| DUT[b], Bruining and van Rooij (1997) | 2 | 5 | 0.9M[c] | 3-hole probe, pressure taps | inverse BEM, stagnation point, probe measurements, frontal pressure taps |
| NREL[b], Simms et al. (1999) | 3 | 5 | 0.7M[c] | wind vane, 5-hole probe, pressure taps | probe measurements, stagnation point normalization, matching up $C_P$, inverse BEM |
| DTU[b], Madsen et al. (1998) | 3 | 9.5 | 1M[c] | 5-hole probe | probe measurements |
| MIE[b], Maeda et al. (2005) | 3 | 5 | 0.5M[c] | 5-hole probe, pressure taps | probe measurements |
| DanAero, (Bak et al., 2011a) | 3 | 40 | 1.5 − 6.1M | 5-hole probe, pressure taps, microphones | probe measurements, matching up $C_P$ |
| **Wind Tunnel** | | | | | |
| MEXICO, Schepers et al. (2012) | 3 | 2.25 | 0.8M[d] | pressure taps | inverse BEM, inverse free wake, based on CFD |
| LMEE, Sicot et al. (2008) | 2 | 0.67 | 300k | pressure taps | lifting line |
| BeRT, Klein et al. (2018) | 3 | 1.5 | 290k | 3-hole probe, pressure taps | probe measurements, based on CFD |
| UW, Moscardi and Johnson (2016) | 3 | 1.7 | 300k | 5-hole probe | probe measurements |

(a) $Re_c$: Reynolds number based on chord length at 70%R and relative inflow velocity. (b) Additional information can be found on the International Energy Agency (IEA) Annexes reported by Schepers et al. (1997) and Schepers et al. (2002). (c) Summarized in the IEA Annexes reported by Schepers et al. (2002). (d) Reported by Schepers and Schreck (2019).

**L46:** it depends on the blade length and scaling; a 20mm pitot tube in comparison with a 60m blade- please adjust this against wind tunnel testing

**Authors answer:** This sentence was rewritten: being explicit that this is relevant in the case of small test turbine models, where this dimensions are comparable:

"In general, according to the published literature, external probes can be used to determine the AoA. However, in the case of wind turbine models, such probes are intrusive and significantly disturb the flow over the blade section where are mounted."

**L48:** no references given

**Authors answer:** This general statement is now followed by several citations regarding each research that employs surface pressure data.

*"Other complementary tools, used on research turbines are surface pressure sensors, located along the blade chord. These sensors are used to record the pressure distribution along the blade chord at a desired radial position and to calculate the aerodynamic loads. Different computational methods use this information has a source to estimate the AoA.*

*The inverse BEM method is probably the most common. From the surface pressure sensors, the normal and tangential forces are calculated. Assuming that they are uniform over an annulus containing the blade section. The wake-induced velocities are calculated according to momentum theory, yielding the effective velocity vector and subsequently the AoA (Whale et al., 1999). This method was implemented by ECN, NREL, DTU projects, obtaining similar results with their respective probes estimations.*

*Schepers et al. (2012) presented the inverse free wake method applied to the MEXICO rotor, which follows the same BEM principle but using the normal and tangential forces into a free wake model. Several computational methods can be found in the latest phase of the project, summarized by Schepers et al. (2018), such as azimuth average, three point and lifting line average methods among others.*

*The surface pressure measurements also allow experimental estimations. Shipley et al. (1995) showed the stagnation point normalization method described as follows: the local dynamic pressure is estimated as the maximum value of the pressure side in each pressure distribution station. This value is used to estimated the freestream velocity and then the AoA based on the geometrical velocities defined by pitch, yaw and azimuth angles.*

*Moreover, Brand (1994) presented the stagnation point method. The AoA is estimated as follows: The stagnation point is located as the previous method. Afterwards, the intersection of the chord line and a line normal to the surface at the stagnation point is used to estimate AoA. The position of the point of intersection can be determined 2D approaches either codes or wind tunnel measurement (Whale et al., 1999). The drawback of this method is that it relies only in the geometry of the blade section, assuming AoA and Reynolds number no influence.*

*Furthermore, Bruining and van Rooij (1997) exposed an additional method that use two frontal pressure taps, one on the pressure side and one on the suction side, working as an built-in probe in the blade. The drawback of this is that requires calibrating the blade station where the taps are located.*

*Schepers et al. (2002) reported the comparison between experimental probes, pressure taps and inverse BEM methods regarding the field measurement from ECN, NREL, DUT, DTU and Mie. The main conclusions found were: (1) The ambiguity of the 3D AoA definition implies that any check on accuracy can only be carried out with an arbitrary reference. (2) Before stall, the estimations of the AoA remain with differences below 1°. (3) Above stall conditions, the differences between methods can go up 4°. Table 1 shows field and wind tunnel experiments with the most common estimation methods mentioned above."*

**L51:** unclear sentence: With a pitot pressure sensor, you know the position geometrically.

**Authors answer:** This paragraph has been removed because the wording was confusing. Probes and pressure taps topics are now addressed in separate parts of the literature to improve the text structure.

**Figure 2** is missing definitions (Ut, Un, Urel), as well as t/n

**Authors answer:** This is now included in the caption on the Figure. Additionally, they are in the list of symbols

[Figure]

**Figure 3.** Angles definition. Azimuth, $\phi$ and yaw, $\psi$ (left). Angle of attack, $\alpha$, pitch, $\theta$ and twist, $\gamma$. $U_t$, $U_n$ and $U_{rel}$ are the tangential, normal and relative velocities, respectively (right).

**L109:** explain 'on a comparable level', e.g what is the implicit effect of Turbulence (1.5%)?

**Authors answer:** This has been included. Additionally, Fig. 2 has been added to show the level of homogeneity of the inflow and the velocity distributions over the four relevant radial positions.

*"With this level of turbulence it can be expected small variations between rotations of the turbine, which suggested includes several rotations in the measurement data.*

*At the same time, the inflow showed some heterogeneity, i.e. was not fully uniform as is depicted in Fig. 2 (left). Figure 2 (right) shows four axial velocity distributions over at the radial positions 45; 65; 75 and 85%R. Therefore, due to these characteristics it was decided to analyze the measurement data over small azimuth angle stations."*

[Figure]

**Figure 2.** Axial inflow. Dashed lines: tip and tower positions. Colored lines: radial positions at 45, 65, 75 and 85%R following the blade rotation (left). Velocity distributions over radial positions at 45, 65, 75 and 85%R (right).

**L118:** The choice for using Clark-Y is not clear (high drag airfoil), see f. example DOI: 10.2514/6.2006-33 Conference: 44th AIAA Aerospace Sciences Meeting and Exhibit.

**Authors answer:** Additional information has been added (see below). Regarding the high drag values in Fig 2 (DOI: 10.2514/6.2006) Cl/Cd plot, these are for very low Reynolds number ($< 10^5$). In the present experiment, under rated conditions, the Reynolds number range is $3 \times 10^5$ for the radial range $1.7 \times 10^5 < Re < 3 \times 10^5$.

*"A slightly modified Clark-Y airfoil profile is used along the entire blade span and there is no cylindrical root section. The airfoil modification was necessary in order to account for a realistic trailing edge thickness with respect to manufacturing requirements. Aerodynamically, the design intended to avoid stall while keep offering optimal performance and the maximum internal space to include instrumentation (Pechlivanoglou et al., 2015).*

*In this way, the specific airfoil profile was chosen as it performs well at low Reynolds number (Re), i.e. at the conditions relevant to BeRT (Re range from 170k to 300k along the span). The blade twist was selected so that the local AoA stays constant over the span at rated conditions"*

**L113:** Model Blockage and consequences for interpretation of results?

**Authors answer:** A model to analyze the blockage effect has been implemented. This is in terms of the equivalent freestream velocity. Additionally, this is coupled with the geometrical calculations. The hypothesis that the offset in AoA, $\Delta\alpha = 2.3°$, between experimental approaches and analytical estimations (Figure 13 of the original submission given below for reference) was a consequence of the blockage is now strengthened by this correction. More changes due to the inclusion of this correction follow in the next answers to the referees' comments. Figure 15 shows the effect of considering the blockage correction on the analytical calculation.

[Figure]

**Figure 13.** AoA results for yaw angle $\psi = -15°$ and pitch angle $\theta = 0°$. Pressure taps and 3-hole probe approaches (left). Analytical calculations (right).

*"The turbine rotor area ($A_{BeRT}$) produces a considerable blockage ratio in the wind tunnel, $\epsilon = A_{BeRT}/A_{tunnel} \approx 0.4$. The blockage effect was analyzed in terms of the equivalent freestream velocity ($U'$) which produces the same torque. Glauert (1926) showed that for a propeller the ratio between the wind tunnel velocity ($U_\infty$) and its corresponding equivalent freestream velocity is a function of the blockage ratio and the thrust coefficient ($C_T$), Eq. 1. Using the BeRT rotor characteristics reported by Marten et al. (2019), a thrust coefficient of $C_T = 0.77$ (expected at rated condition) was considered. Subsequently, applying Eq. 1, implemented on wind turbines, results in the velocity ratio of $U_\infty/U'=0.86$.*

$$\frac{U_\infty}{U'} = \left(1 - \frac{\epsilon C_T}{4\sqrt{1 + C_T}}\right)^{-1}$$

*It is noted that this correction has also been applied successfully in wind tunnel experiments with even higher blockage ratio (45% Refan and Hangan, 2012)"*

[Figure]

**Figure 15.** AoA results for yaw angle $\psi = -15°$ and pitch angle $\theta = 0°$. Pressure taps and 3-hole probe approaches (left). Analytical calculations (right).

**L115:** is the turbine yaw fixed or free?

**Authors answer:** Both the turbine yaw angle and the blade pitch angle are fixed during the measurements. The text has been rewritten:

*"The turbine yaw angle and the blade pitch angle were fixed during the measurements"*

**L124:** the statement of placement of pressure taps is not constant=0.45-why straight line placement?/why is it in this small scale experiment not following constant radius?

**Authors answer:** The reason is related to have good comparability with 2D airfoil studies. A drawing of the pressure taps together with the curvature (red line) is shown here. Although the curvature error is considered small ($\Delta r$<0.025 $m$), it was considered when the pressure was corrected by centrifugal effect as shown Eq 2.

$$P_{corr} = P_{si} + \frac{\rho}{2}(\Omega r_i)^2$$

[Figure]

**L126:** what is the max frequency (3 dB limit) of the detectable signal. **L129** specs?

**Authors answer:** Spectra from both experimental tools (3-hole probes and pressure tap) have been included in Fig. 7.The same spectra shown in Figure 7 is presented here in [dB], the 3dB line is plotted in red with the max frequency at $\approx 6Hz$. More information regarding the filtering decisions using the spectra can be found in the following answers.

[Figure]

**Figure 7.** Frequency spectrum of one pressure sensor of the 3-hole at 75%$R$ (left). Frequency spectrum of the pressure tap at $x = 2\%c$ (right). Both cases on pitch angle $\theta = 0°$ and yaw angle $0°$

**L134:** A miss why the use of flaps with consensus on title /intro & science objectives

**Authors answer:** The TE-flaps were set in their neutral position for all the experiments, and they are given in the description only for completeness of the blade information. The sentence below explicitly states this. Additionally, the same information has been added to the caption of Figure 5.

*"The flaps were fixed without any deflection for all test cases presented in this study."*

[Figure]

**Figure 6.** 3-hole probes mounted in the equipped blade (left). Calibration of a 3-hole probe and tip details (right). It is noted that although the flaps appear deflected in the photo, they were always in the neutral position for the experiments of this campaign.

**L157:** using a 3 hole probe-no side slip detection. What about the flow conditions when the turbine is in yaw? **L237:** the discussion of cross flow (sideslip) for the 2D probe is missing. Or may be your statement is to use a 2D probe in the 3D inflow as a representation of the normal(tangential) velocity components? Clarification and error calculation is needed. **L310:** the question is if yaw affetcs the pressure in the dynamic inflow field, observed here with a 2D-probe.

**Authors answer:** The effectiveness of the 2D probe over the misaligned cases has been addressed and is added here for convenience:

*"As the turbine was set under yaw misalignments, it is important to verify the effectiveness of the 2D probe. The range of the AoA, in the probe stations, is $0° \leq \alpha \leq 10°$. Therefore, adding the corresponding twist angle, the range of the AoA relative to the probes $\alpha_{probe} \leq 18°$. Moreover, the probes are aligned with the chord, thus the yaw angle relative to the probe is the same, $-30° \leq \psi_{probe} \leq 0°$.*

*Zilliac (1993) and Moscardi and Johnson (2016) determined the mono-zone as $\pm 30°$ $(\alpha_{probe}, \psi_{probe})$. This zone represents where the calibration parameters of the probes remain invariant, i.e. $C_{P,probe}$. These studies used probes with 7- and 5-holes, respectively. As a 3-hole probe sweep the same angle of these calibrations, its monozone should be the same.*

*Moreover, Bruining and van Rooij (1997) employed 3-hole probes on field measurements with good agreement of the AOA, compared to inverse BEM and stagnation point methods. In addition, Klein et al. (2018) showed similar results from experimental and CFD simulations where the wind tunnel structure was considered. Therefore, based on these arguments, it was assumed that the 3-hole probes are able to estimate the AoA in the yaw misalignments here studied."*

Figure 6a from Zilliac (1993) is presented here, highlighting the zone of misalignment where the current experiments were made.

[Figure]

(Disclaimer: This picture does not belong to any of the authors and is presented here only as a support of the statement. Author reference: Zilliac, G.: Modelling, calibration, and error analysis of seven-hole pressure probes, Experiments in Fluids, 14, 104–120, 1993.)

**L171:** what is the explanation behind seeing the 1P in the signal for the interpretation?

**Authors answer:** The exact cause of the 1P frequency in the signal is unconfirmed at the moment. It is conceivable that it is caused by either some rotor imbalance or by the tower effect or both.

**L174:** This is a surprising statement about resonator box system that doesn't damp frequencies.30 Hz filter? The cited reference (Berg) offers fig 21(assuming small tubes) with considerable amplitude and phase lag properties. This needs clarification

**Authors answer:** The authors realize that the 30Hz filter was unsuitable, as it was higher than the structural frequencies of the blade and rotor ($f_{blade} \geq 13.5\ Hz, f_{tower} \geq 18Hz$) and also for dynamic response purposes, as the reviewer rightly highlights.

*"The structural design of BeRT results in eigenfrequencies of the blades $f_{blade} \geq 13.5\ Hz$ and the tower $f_{tower} \geq 18Hz$. For this reason, the data were low pass filtered using a Butterworth filter with a cut off frequency of $12Hz$ to reduce the noise and structural vibrations."*

*"The dynamic response of the pressure taps/tubes system was evaluated theoretically following the model proposed by Bergh and Tijdeman (1965). Figure 8 (left) shows a scheme of the model used to apply the analysis, based on the tube arrangement depicted in Fig. 5, while Figure 8 (right) shows the theoretical response of the system, based on Bergh and Tijdeman (1965). In order to minimize the attenuation and phase lag of the signal, an additional low pass filter was applied, with a cut off frequency of $6Hz$. This was considered adequate as it shows the amplitude amplification and phase lag are less than 1% and 10°, respectively."*

[Figure]

**Figure 8.** Scheme of the model to apply Bergh and Tijdeman (1965) dynamic response analysis, $P$, $l$ and $d$ are the pressure, length and diameter of each section (left). Theoretical dynamic response of the amplitude and phase lag (right).

To provide further insight, an additional figure is given below, where the effect of different filters is shown. Figure AA1 (left) shows the previous version, where only a low pass filter with a cut off frequency of 30Hz was used. Figure AA1 (right) shows the current results, a cut off frequency of 12 Hz in the case of the 3-hole probes and a lower cut off frequency (6Hz) for the pressure taps, in order to avoid large phase lag and damping. The two measurement tools present an improvement, reducing the vibrations and resulting in a smoother behavior.

[Figure]

Figure AA1: AoA results from pressure taps and 3-hole probes approaches. Low pass filter with cut off frequency of 30Hz (left). Low pass filter with a cut off frequency of 12Hz and 6Hz over the 3-hole probes and pressure taps, respectively.

**Figure 10**: What is the difference between the black-dashed and red points around x/c=0.3..and onwards?

**Authors answer:** This has been included.

"The difference between both curves $\Delta C_P \leq 0.05$ until $x = 30\%c$, except the peak at the suction side ($\Delta C_P(x = 1\%c) = 0.2$). Afterwards, $\Delta C_P$ varies between $0.05 - 0.10$."

**L253:** temperature increases in the flow during experiments effects on the pressure sensors (standard calibration at 25deg nom)? As I recall the HCL's have +-0.25%FS nonlinearity & hysteresis. So i would assume higher uncertainty on aoa. Table1 needs to state that uncertainty is [fraction/%] of FSR

**Authors answer:** According to the manufacturer, the $\pm 0.25\% FS$ is the maximum, the nominal value is $\pm 0.05\% FS$. During the experiments, the temperature range was 17.5-19.5°C. The nominal value was considered to calculate the errors.

*"During the measurement campaign, between test cases, the tunnel was opened, meanwhile the changes on the pitch or yaw angle were made. This allowed to keep the temperature and relative humidity within range of $18 \pm 5°C$ and $40 \pm 5\%$, respectively."*

Subsequently, the error calculation includes the nominal values, phase standard deviation and the conversion from pressure to AoA:

*"The measurement uncertainty, for of all quantities, was taken into account in order to quantify the error magnitude over the results. Both AoA estimation approaches have the same iteration in the error propagation, based in the following steps:*

*1. Nominal error of each sensor.*

*2. The measurement standard deviation of the averaged measurements. This was calculated with the same azimuth step as the phase average.*

*3. Conversion to AoA. Thus, the error propagation after applying Eqs. 3 and 5 for the 3-hole probes and pressure taps, respectively.*

*Table 2 shows the overall uncertainty for all the quantities. The point 3. depends highly on the values of the measured pressure. For this reason, Table 2 shows the minimum and maximum values. An example of the uncertainty over the azimuth angle of each tool can be seen in App. D1"*

It was decided to keep the units in the table to provide the uncertainty of the pressure sensors and AoAs together. An additional figure was added in the appendix (Fig D1), showing the uncertainties over each measurement tool along the azimuth angle in order to visualize their level. Here is for convenience

[Figure]

**Figure D1.** AoA results from the pressure tap and 3-hole probe approaches with their uncertainties. Pitch angles: $\theta = 0°$ and yaw angle $\psi = -30°$.

The figures from the section 4 of the Results were redone following the changes detailed above. The changes are regarding the new filtering

**L266:** The results are expressed in Pascal, may be it is more clear to show it relative (normalisation), speaking of uncertainty and also from a point of measurement range.

**Authors answer:** The pressure values were normalized by the dynamic pressure inflow. Consequently, the results are now non dimensional. The uncertainties described before are included in the plot and the measurement range and the level of the uncertainty on those ranges.

[Figure]

*"In terms of the measurement range, the relative pressure $2.8 \leq q_{rel}/q_\infty \leq 6.5$. Over this range, the uncertainty error represents the $4.5\%$. In the case of the pressure difference at $12.5\%c$, the range is $6 \leq \Delta P(12.5\%c)/q_\infty \leq 10.3$, where the error takes a value of 4%."*

**Figure 14:** add of result for 0 yaw missing

**Authors answer:** This has been included:

*"Figure 13 shows the relative dynamic pressure at the radial position $r = 45\%R$ for the aligned and misaligneds cases, normalized by the dynamic pressure $q_\infty$. It can be seen the same trend between the geometrical case (dashed line) and the estimation from the pressure taps (PP, solid line) as well in the maximum ($\phi = 0°$) and minimum ($\phi = 180°$) azimuth positions."*

[Figure]

**Figure 13.** Normalized relative dynamic pressure at radial position $r = 45\%R$ and yaw misalignment cases of $\psi = -15°$ and $\psi = -30°$. Solid line, pressure tap estimation. Dashed line, geometrical calculation.

**Figure 12** Odd, with the 2-2½P variations(L316), except for the tower influence... Check! **L334** Could this be the damping effects from the resonating tubes characteristics?, same p variation issue as above

**Authors answer:** The addition of the new filters and the inflow heterogeneity explain the behavior.

*"The explanation is due to the heterogeneity of the inflow. These variations, $\Delta U_\infty = \pm 0.2 ms^{-1}$ (see Fig. 2), can have the same influence that the tower over the AoA estimations. Using the geometrical estimation, $\alpha_{geo}$, under this level of variation in the inflow, results in AoA difference of $\Delta \alpha_{geo} = \pm 0.4°$, which support this statement."*

[Figure]

The main reason was that the filter decision in the first version, did not count the lower Eigen frequencies of the blade and tower. Also with the new filter, it seems that the vibration on the mounting of the 3-hole probes was suppressed.

**Referee #2**

**L139:** Is there a reason for this very high logging frequency?

**Authors answer:** The experimental set up has been designed to fulfill the requirements of active flow control with flaps. This information has not been included in the text as it is not relevant to the objective of this study.

**L148:** Check how the references are referenced

**Authors answer:** The references are now between parentheses.

*"According to the BeRT design specification, the combination of chord and twist distribution achieves an optimal shape (Pechlivanoglou et al., 2015) which provides a constant AoA over most of the blade span (Bartholomay et al., 2017), so the AoA at the radial position of the pressure taps and the 3-hole probes should be the same under aligned flow conditions."*

**L187:** Can this fit be shown and has the fit been checked with measurements from the blade at standstill? If this is possible?

**Authors answer:** An example of the downwash correction has been added in Eq. 4:

*"Equation 4 shows an approximation of the downwash correction (Klein et al., 2018).*

$$\alpha = 0.58°\alpha_{probe} - 0.64°$$

*"*

**L200:** Can you elaborate a little more on this? Why is it neglected and what (if any) consequences does it have?

**Authors answer:** Including these terms would imply additional added masses. Nevertheless, the quasi-static condition of evaluation that was applied is found enough to neglect this degree of freedom.

**L227:** It would be good with a comment on how good the fit is or perhaps a figure that shows the fit.

**Authors answer:** The coefficient of determination has been included:

*"The fit values are $k_1 = 0.23$ and $k_2 = 0.43$, with a coefficient of determination of $R^2 \geq 0.999$."*

**L231:** Why did you choose this position and not one where there were pressure taps, so you can avoid the interpolation?

**Authors answer:** The intention is to perform Gaunaa's method with as little modification as possible. The literature and the applied assumptions are related to this position. The introduction of alternative positions would imply a re-evaluation of all the conditions described Section 3.3.2, or create confusion on potential users. Thus, with the idea of the implementation of the method in other test rigs, it was decided to keep it.

**L247:** Change "Equation 7 it can..." to "Equation 7 can..."

**Authors answer:** Done.

**Table 2** What is "FSR"?

**Authors answer:** FSR stands for Full Scale Range. It was changed to "Range"

**L274:** Rephrase the sentence, especially the part with "...to after drops..."

**Authors answer:** This sentence has been rewritten.

*"The pressure difference at $12.5\backslash\%c$ remains relatively constant, $\Delta P(12.5\backslash\%c) = 9.8\ q_\infty$, until the azimuth angle $\phi = 90°$, to afterwards, decreasing continuously until the azimuth $\phi = 180°$ where it reaches its minimum, $\Delta P(12.5\backslash\%c) = 9.3\ q_\infty.$"*

**L297:** Rephrase the sentence. It is an important observation, but difficult to understand as it is written here.

**Authors answer:** This sentence has been rewritten.

*"The pressure taps are located at discrete points on the blade surface. For this reason, the sensor that estimates the stagnation point, i.e. the values of the relative dynamic pressure, fluctuate in location. The latter explains the sharp changes present in yaw angle $\psi = -15°$ at azimuth angles $\phi \approx 70°$ and $\phi \approx 300°$ and yaw angle $\psi = -30°$ at azimuth angles of $\phi \approx 50°$ and $\phi\approx 320°$ (see Fig. 11)."*

**L303:** List of symbols Missing expression for the tip speed ratio.

**Authors answer:** The tip speed ratio is included in the list of symbols. Also in the text, it was defined in section 3.4.

"For all cases, the tip speed ratio was fixed $\lambda = 4.35$."

**Figure 15** Legend is different from Figures 16, 17 and 18. Figure 15 Perhaps repeat that AoAgeo = 5.1 deg from.

**Authors answer:** Done. The Analytical AoA estimation was included, to keep the same format of the misalignment cases (with the blockage correction. 5.1deg is no longer valid).

**L321:** Can the uncertainties be added on the figure?

**Authors answer:** This would overload the Figures at the section. Nevertheless, One representative Figure that includes the uncertainties is now included in the appendix, Fig D1.

[Figure]

**L322-23:** Rephrase

**Authors answer:** This sentence has been rewritten.

*"Although the AoA over the azimuthal variation is not constant, both methods estimate a similar AoA range. The AoA for both pressure tap and 3-hole probe methods are slightly lower than previous experimental results show by Klein et al. (2018), but within the uncertainty values. Table 3 shows the range ($\alpha_{min}, \alpha_{max}$) and average ($\bar{\alpha}$) values of the AoA over the azimuth angle for the pressure taps and the 3-hole probe methods. The range of the tools measurements is between 6.6-7.8° and the geometrical estimation between 6.4-6.8°"*

**L324-26:** Can the effect of the walls be quantified, e.g. by the method from H. Glauert. The elements of aerofoil and airscrew theory? Rephrase

**Authors answer:** A model to analyze the blockage effect has been implemented. This is in terms of the equivalent freestream velocity. Additionally, this is coupled with the geometrical calculations. The hypothesis that the offset in AoA, $\Delta\alpha = 2.3°$, between experimental approaches and analytical estimations (Figure 13 of the original submission given below for reference) was a consequence of the blockage is now strengthened by this correction. More changes due to the inclusion of this correction follow in the next answers to the referees' comments. Figure 15 shows the effect of considering the blockage correction on the analytical calculation.

[Figure]

**Figure 13.** AoA results for yaw angle $\psi = -15°$ and pitch angle $\theta = 0°$. Pressure taps and 3-hole probe approaches (left). Analytical calculations (right).

*"The turbine rotor area ($A_{BeRT}$) produces a considerable blockage ratio in the wind tunnel, $\epsilon = A_{BeRT}/A_{tunnel} \approx 0.4$. The blockage effect was analyzed in terms of the equivalent freestream velocity ($U'$) which produces the same torque. Glauert (1926) showed that for a propeller the ratio between the wind tunnel velocity ($U_\infty$) and its corresponding equivalent freestream velocity is a function of the blockage ratio and the thrust coefficient ($C_T$), Eq. 1. Using the BeRT rotor characteristics reported by Marten et al. (2019), a thrust coefficient of $C_T = 0.77$ (expected at rated condition) was considered. Subsequently, applying Eq. 1, implemented on wind turbines, results in the velocity ratio of $U_\infty/U'=0.86$.*

$$\frac{U_\infty}{U'} = \left(1 - \frac{\epsilon C_T}{4\sqrt{1 + C_T}}\right)^{-1}$$

*It is noted that this correction has also been applied successfully in wind tunnel experiments with even higher blockage ratio (45% Refan and Hangan, 2012)"*

[Figure]

**Figure 15.** AoA results for yaw angle $\psi = -15°$ and pitch angle $\theta = 0°$. Pressure taps and 3-hole probe approaches (left). Analytical calculations (right).

Additionally, The method is coupled with the geometrical calculations. The hypothesis that the offset in the previous version was a consequence of the blockage was now strengthened by this correction:

*"The blockage effect must be considered. Consequently, the inflow velocity ($U_\infty$) for these calculations was replaced by the equivalent freestream velocity. Thus, applying Eq. 1 results in the equivalent freestream velocity of $U' = 7.5ms^{-1}$."*

**L330-31:** Rephrase

**Authors answer:** This sentence has been rewritten.

*"However, the difference is in the same magnitude of the fluctuations of each tool"*

**L353-56:** The paragraph is unclear, please rephrase

**Authors answer:** This sentence was rewritten. The blockage effect was now considered, thus part of the sentence was deleted.

*"The analytical AoAs, Fig. 16 (right), show the same features, including the large difference at azimuth angles $\phi = 0°$ and $\phi = 180°$"*

**L363:** Is there an explanation on the difference in slope? Why is the measurements not showing a slope of -1? **L364** Rephrase

**Authors answer:** This has been included:

*"Nevertheless, the induction factors change at each pitch angle, therefore the change in the slope is the results of that dependency. This agrees with the fact that the slopes are similar but not the same, as is expected variations of the induction factor along the radial positions."*

---

## Editor Decision (ED1)

**Overall comments**
- The overall impact of the paper is not well developed and the scientific novelty not clearly presented nor argued well. This is critical to justifying publication of the article in a scientific journal
- Written very well, but there are still several typos throughout. I have noted those I found but please do a final thorough read through and edit prior to final submission

**Abstract**
- The abstract doesnt speak explicitly to the value of the contribution – what is the potential impact?

**Introduction**
- Typo line 53 should be as not has
- Typo 57 probes' not probes
- Introduction starts nicely and lays out the complexity of the problem. The literature review is very thorough
- However, the literature review and introduction fail to make the case for the unique contribution of this work. What is unique about the approach that makes is journal publication worthy? What is special about the BeRT blade? It still is unclear at the end of the introduction what the point of this paper is from the perspective of scientific contribution.

**Methodology**
- This seems to be the first introduction of the point of this paper on line 181 – why is this a significant contribution? Elaborate!
  - Also, Why is this intent not discussed at the beginning of the introduction? You need to reiterate the point of the paper in the abstract, introduction and in the conclusions. Don't expect the reader to dig it out for themselves
- Why were the particular locations along the blade span chosen and why pressure tabs and 3HP at the respective locations? I don't think again the motivation for why everything is set up as it is is fully developed. The section is more descriptive than analytical
- Line 182 has a typo, either when it is or when applied but not when is
- Typo line 202, while the pressure tap uses the static pressure in…
- Line 227 typo, because the blade itself induces…
- Line 324 to keep the

**Results and discussion**
- Long sentence line 334 is hard to read
- Figure 14 seems to have incomplete caption. Should explicitly say what each figure represents in the caption not just the text
- Typo line 384
- The discussion and interpretation of the differences between the results and to prior publications could be more fully developed – and perhaps help provide the paper

with more analytical substance. Right now, the results section reads very descriptive rather than identifying and pulling out key insights

Conclusions
- Finally in lines 459-470 the paper gets to the point. This was not obvious throughout the paper. Please think about how to strengthen the paper in lifting up and highlighting the novel contributions.

---

## Author Response (AR2)

**Determination of the Angle of Attack on a Research Wind Turbine Rotor Blade Using Surface Pressure Measurements**

Reply to Handling Associate Editor Dr. K. Dykes

Dear Dr. Dykes, thank you for the detailed review of our paper. The comments are addressed in the final version. Please find below, the answers to your comments.
The supplementary documents attached are:

- manuscript_wes-2020-35.pdf: is the last version of the paper
- manuscript_wes-2020-35_tracked_changes.pdf: contains additions in blue and deleted text in strike-through red.

Abstract
- The abstract doesnt speak explicitly to the value of the contribution – what is the potential impact?
R: The abstract was rewritten and is given below for convenience.

"In this paper, a method to determine the angle of attack on a wind turbine rotor blade using a chordwise pressure distribution measurement was applied. The approach used a reduced number of pressure tap data located close to the blade leading edge. The results were compared with the measurements from three external probes mounted on the blade at different radial positions and with analytical calculations. Both experimental approaches used in this study are based on the 2-D flow assumption; the pressure tap method is an application of the thin airfoil theory, while the probe method applies geometrical and induction corrections to the measurement data.

The experiments were conducted in the wind tunnel at the Hermann Föttinger Institut of the Technische Unversität Berlin. The research turbine is a three-bladed upwind horizontal axis wind turbine model with a rotor diameter of 3m. The measurements were carried out at rated conditions with a tip speed ratio of 4.35 and different yaw and pitch angles were tested in order to compare the approaches over a wide range of conditions.

Results show that the pressure tap method is suitable and provides a similar angle of attack to the external probe measurements as well as the analytical calculations. This is a significant step for the experimental determination of the local angle of attack, as it eliminates the need for external probes, which affect the flow over the blade and require additional calibration."

Introduction

- Typo line 53 should be as not has
R. The typo was changed. The sentences is given below for convenience.

"Different computational methods use this information as a source to estimate the AoA".

- Typo 57 probes' not probes
R: The typo was changed. The sentences is given below for convenience.

"This method was implemented by ECN, NREL, DTU projects, obtaining similar results with their respective estimations based on probes".

Methodology

- Why were the particular locations along the blade span chosen and why pressure tabs and 3HP at the respective locations? I don't think again the motivation for why everything is set up as it is is fully developed. The section is more descriptive than analytical
R: The information has been added in the methodology section and is given here for convenience.

"Due to manufacturing reasons (internal structure, holes spacing), the pressure taps could only be located at a single spanwise location, which was at 45\% of the blade span. Each pressure tap was connected through silicone tubes inside the blade to a pressure box located in the hub which contains all sensors."

"The blade was also provided with three trailing edge flaps with 10\%R span length and 30\%c chord length and located consecutively from 60\% to 90\% along the span. Each 3-hole probe was aimed to give feedback information to choose flap movements. However, the flaps were fixed without any deflection for all test cases presented in this study."

- Line 182 has a typo, either when it is or when applied but not when is
R: The typo was changed. The sentences is given below for convenience.

"The main idea is to compare the results obtained by the method proposed by Gaunaa and Anderson (2009) when it is applied to the pressure tap data against the AoA from 3-hole probe measurements and analytical calculations."

- Typo line 202, while the pressure tap uses the static pressure in…
R: The typo was changed. The sentences is given below for convenience.

"The pressure sensors measure the differential pressure $P_{si}$. The 3-hole probes use the inner tube as a reference, while the pressure taps use the static pressure in the test section".

- Line 227 typo, because the blade itself induces…
R: The typo was changed. The sentences is given below for convenience.

"The latter angle differs from $\alpha$, which is the effective AoA of the blade section, because the blade itself induces a velocity on its surroundings".

- Line 324 to keep the
R: The sentence was changed and is given below for convenience.

"During the measurement campaign, while the changes on the pitch or yaw angle were made between test cases, the tunnel was left open to allow for fresh air to enter the tunnel circuit. As a result, the temperature and relative humidity were kept within $18 \pm 1.5°C$ and $40 \pm 5\%$, respectively."

Results and discussion
- Long sentence line 334 is hard to read.
R: This paragraph was rewritten:

"For the aligned case, $\psi = 0°$, the relative dynamic pressure remains relatively constant at $q_{rel} = 4.5q_\infty$, while the pressure difference at 12.5%c exhibits four marked behaviors :

Initially, $0° \leq \phi \leq 90°$, it remains relatively constant at $\Delta P(12.5\%c) = 9.8q_\infty$. Then the dynamic pressure drops, to reach a minimum at $\phi = 180°$ ($9.3q_\infty$), while an increase follows from $\phi = 180°$ to $\phi = 290°$. At that point, the dynamic pressure reaches its maximum value ($10.3q_\infty$) before it starts dropping to reach $9.8q_\infty$ at $\phi = 360°$."

- Figure 14 seems to have incomplete caption. Should explicitly say what each figure represents in the caption not just the text
R: Yes, thank you for highlighting this. After the first revision, the figure on the right was added without updating the caption. This has been corrected and the figure is given below for your convenience

[Figure]

**Figure 14.** AoA results for yaw angle $\psi = 0°$ and pitch angle $\theta = 0°$. Pressure taps and 3-hole probe approaches (left). Analytical calculations (right).

- Typo line 384

R: The sentence was rewritten:

"These variations, $\Delta U_\infty = \pm 0.2 ms^{-1}$ (see Fig. 2), can have the same influence as the tower over the AoA estimations. The geometrical estimation ($\alpha_{geo}$) under such inflow variations, results in an AoA difference of $\Delta \alpha_{geo} = \pm 0.4°$, which supports this statement"

- The discussion and interpretation of the differences between the results and to prior publications could be more fully developed – and perhaps help provide the paper with more analytical substance. Right now, the results section reads very descriptive rather than identifying and pulling out key insights

R: The results are discussed in more detail and in an analytical manner in the sections listed below. In the interest of brevity, we consider this is suitable for this paper.

Lines 359-371: Explain with details the effect of the discrete taps in the chordwise surface.
Lines 391-394: Explain and highlight the fact of the non-uniform inflow incidence.
Lines 399-401: Evidence that the differences of previous studies is the blockage, as the model implemented in the analytical approach includes that effect.
Lines 441-444: Explain why the pitch angle and angle of attack do not present a one to one linear relation, highlighting the inherent axial induction factor dependency.

-On Introduction: starts nicely and lays out the complexity of the problem. The literature review is very thorough. However, the literature review and introduction fail to make the case for the unique contribution of this work. What is unique about the approach that makes is journal publication worthy? What is special about the BeRT blade? It still is unclear at the end of the introduction what the point of this paper is from the perspective of scientific contribution.

-On Methodology: This seems to be the first introduction of the point of this paper on line 181 why is this a significant contribution? Elaborate!  Also, Why is this intent not discussed at the beginning of the introduction? You need to reiterate the point of the paper in the abstract, introduction and in the conclusions. Don't expect the reader to dig it out for themselves

On Conclusions: Finally in lines 459-470 the paper gets to the point. This was not obvious throughout the paper. Please think about how to strengthen the paper in lifting up and highlighting the novel contributions.

R: The suggestion of the paragraphs of the methodology has been taking into account, now the impact and contribution is explicitly in the abstract, introduction and conclusions.

Abstract:

"[…]. Results show that the pressure tap method is suitable and provides a similar angle of attack to the external probe measurements as well as the analytical calculations. This is a significant step for the experimental determination of the local angle of attack, as it eliminates the need for external probes, which affect the flow over the blade and require additional calibration."

Introduction:

"[…]. To the authors' knowledge, this method has not been applied on a rotating blade yet. Given the good agreement between 2-D and 3-D pressure distributions away from the root region, this paper presents an alternative method of determining the AoA by means of pressure tap measurements. The present investigation aims at providing experimental verification for one such surface pressure method (Gaunaa and Anderson 2009) on the rotating blade".

Conclusions:

"[…] Overall, it is found that the pressure tap method applied here to determine the AoA, provides reliable data, with good performance for both aligned and misaligned cases. Hence, the presented method is a promising alternative to the use of external probes, which affect the flow over the blade and require additional calibration".